# VisPhyWorld: Probing Physical Reasoning via Code-Driven Video Reconstruction

## Abstract

Evaluating whether Multimodal Large Language Models (MLLMs) genuinely reason about physical dynamics remains challenging. Most existing benchmarks rely on recognition-style protocols such as Visual Question Answering (VQA) and Violation of Expectation (VoE), which can often be answered without committing to an explicit, testable physical hypothesis. We propose **VisPhyWorld**, an execution-based framework that evaluates physical reasoning by requiring models to generate executable simulator code from visual observations. By producing runnable code, the inferred world representation is directly inspectable, editable, and falsifiable. This separates physical reasoning from rendering. Building on this framework, we introduce **VisPhyBench**, comprising 209 evaluation scenes derived from 108 physical templates and a systematic protocol that evaluates how well models reconstruct appearance and reproduce physically plausible motion. Our pipeline produces valid reconstructed videos in 97.7% on the benchmark. Experiments show that while state-of-the-art MLLMs achieve strong semantic scene understanding, they struggle to accurately infer physical parameters and to simulate consistent physical dynamics.

## 1. Introduction

Recent advances in Multimodal Large Language Models (MLLM) have led to impressive performance on a wide range of visual and language tasks (Fu et al., 2024). However, assessing whether such models exhibit principled physical reasoning remains challenging. Existing evaluation protocols often rely on recognition-based queries or surface-level judgments, which can obscure whether correct outputs arise from coherent physical reasoning or from

[1]Anonymous Institution, Anonymous City, Anonymous Region, Anonymous Country. Correspondence to: Anonymous Author <anon.email@domain.com>.

Preliminary work. Under review by the International Conference on Machine Learning (ICML). Do not distribute.

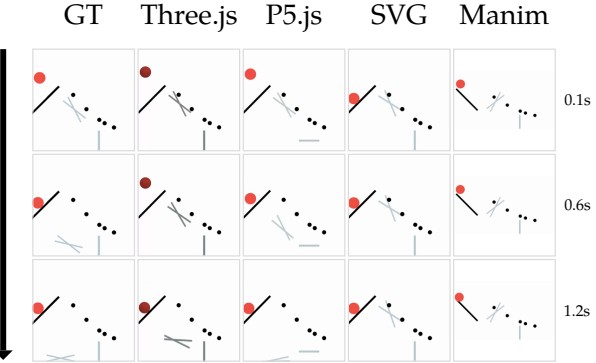

GT    Three.js    P5.js    SVG    Manim

0.1s

0.6s

1.2s

*Figure 1.* **MLLMs struggle to simulate physical dynamics.** Under the same inputs, code generated with rigid-body simulation backends (Three.js/P5.js) produces more physically consistent rollouts, whereas non-physics backends (SVG/Manim) often exhibit implausible motion or contact artifacts such as interpenetration.

learned visual correlations (Chen et al., 2023; Shen et al., 2025). Most benchmarks probe physical understanding through passive recognition tasks such as Visual Question Answer- ing (VQA)-style or Violation of Expectation (VoE)-inspired recognition tasks (e.g. CLEVRER (Yi et al., 2020), GRASP (Jassim et al., 2024), MVPBench (Krojer et al., 2025))). These settings can reward dataset-driven guessing, encouraging memorized priors and surface-level pattern matching rather than genuine causal physical understanding (Pezeshkpour & Hruschka, 2023; Keluskar et al., 2024). This challenge is particularly acute for MLLMs, which typically output only text and therefore do not provide predictive likelihoods or measures of surprise commonly used to evaluate generative world models (Garrido et al., 2025). We therefore argue that in this context, **evaluation should require reconstruction and re-simulation, forcing models to commit to an explicit physical visuals rather than merely select an answer or text reasoning.** We propose VisPhyWorld, a paradigm shift: using executable code as a test of physical understanding, as illustrated in Figure 2. VisPhyWorld probes the physical reasoning capabilities of MLLMs through visual-to-code reconstruction. Given two key frames (and optionally object detections), the model produces executable simulation code that recreates the scene and rolls it forward to synthesize future frames as shown in

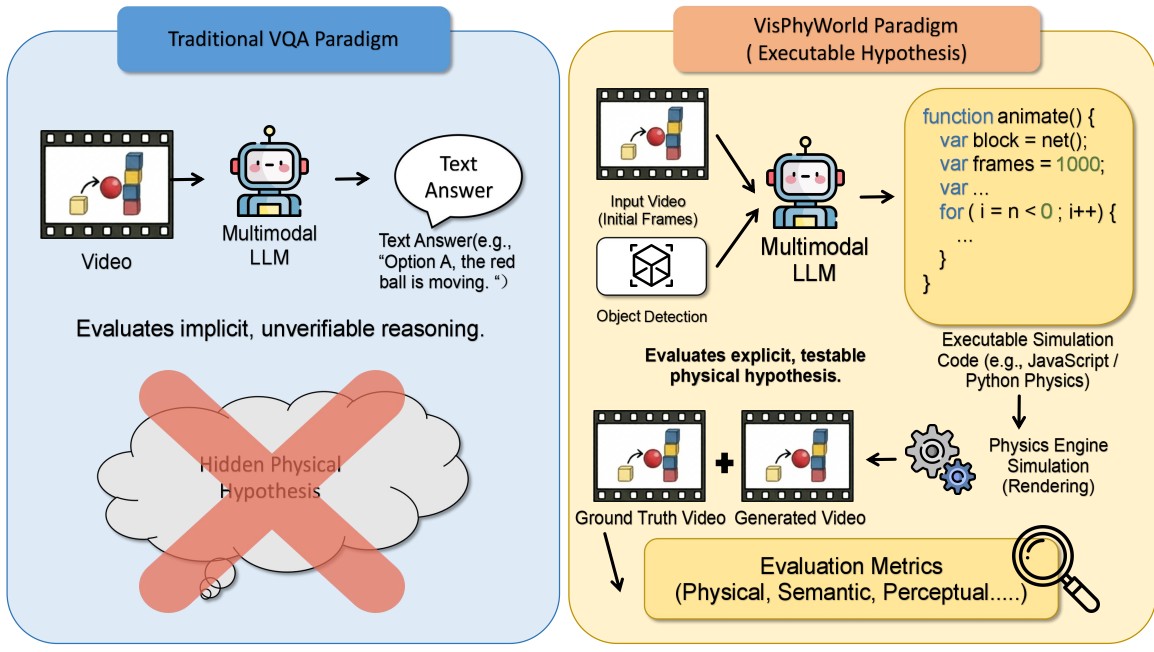

*Figure 2.* Unlike traditional VQA paradigms, **VisPhyWorld accesses physical understanding evaluation** by requiring MLLMs to actively reconstruct scenes via executable code, offering superior reasoning explainability compared to traditional paradigms.

Figure 3. This process not only produces the video but does so in a fully interpretable and editable manner. Beyond the rendered video, VisPhyWorld exposes the generated code itself as a reasoning artifact, making the model's physical logic directly inspectable.

We also introduce VisPhyBench, a standardized evaluation suite with a systematic protocol that assesses how well models reconstruct appearance and reproduce physically plausible motion across complementary perspectives. Our investigation reveals a critical insight: while current state-of-the-art LLMs excel at semantic recognition, they exhibit significant limitations in fine-grained physical comprehension, often failing to parameterize simple Newtonian dynamics correctly even in a simple 2D setting, let alone in 3D environments. In summary, our contributions are threefold:

(1) We propose **VisPhyWorld**, a framework that uses LLMs to interpret raw video frames and generate executable simulation code for predicting future motion. To our knowledge, this is the first paradigm that evaluates physical reasoning in MLLMs through code reconstruction and re-simulation. By making object states and dynamics explicit, VisPhyWorld provides a direct and interpretable view of a model's physical understanding.

(2) We introduce **VisPhyBench**, a unified evaluation protocol comprising 209 scenes derived from 108 physical templates that assesses physical understanding through the lens of code-driven resimulation in both 2D and 3D scenes, integrating metrics from different aspects.

(3) We provide an in-depth analysis of current MLLM, demonstrating that despite their linguistic prowess, they

**fail to grasp the fundamental dynamics of real-world motion**. Our results reveal a critical gap: while models can accurately describe scene contents, they struggled to reconstruct the scene in a way that conformed to the laws of physics, indicating that they rely on superficial visual pattern matching rather than a grounded understanding of physical causality.

*Table 1.* VisPhyWorld uniquely turns physical reasoning into an executable hypothesis and enables multimetric, diagnostic evaluation beyond relative scoring.

| Benchmark | Future Visual Generation | Evaluates MLLM Outputs | Executable Hypothesis | Evaluation Paradigm |
|---|---|---|---|---|
| PHYRE (Bakhtin et al., 2019) | ✗ | ✗ | ✗ | Relative (Actions) |
| CLEVRER (Yi et al., 2020) | ✓ | ✗ | ✗ | Relative (QA) |
| IntPhys (Riochet et al., 2020) | ✓ | ✗ | ✗ | Relative (VoE) |
| PhyGenBench (Meng et al., 2024) | ✓ | ✓ | ✗ | Relative (QA) |
| MVP (Krojer et al., 2025) | ✗ | ✓ | ✗ | Relative (QA) |
| PhysicsIQ (Motamed et al., 2025b) | ✓ | ✓ | ✗ | Relative (QA) |
| WorldModelBench (Li et al., 2025) | ✓ | ✓ | ✗ | Absolute (VLM Judge) |
| IntPhys2 (Bordes et al., 2025) | ✗ | ✓ | ✗ | Relative (VoE) |
| PhyWorld (Kang et al., 2025) | ✓ | ✗ | ✗ | Reconstruction (Video) |
| **VisPhyWorld (Ours)** | ✓ | ✓ | ✓ | **Reconstruction (Code)** |

## 2. Related Work

**Intuitive physics.** Understanding the world is commonly studied through physical reasoning tasks that probe models' ability to infer object dynamics, interactions, and causal relationships from visual input (Melnik et al., 2023; Fung et al., 2025). Inspired by findings from developmental psychology showing that infants exhibit sensitivity to physical violations (Baillargeon et al., 1985), prior work on intuitive physics investigates whether models can anticipate physi-

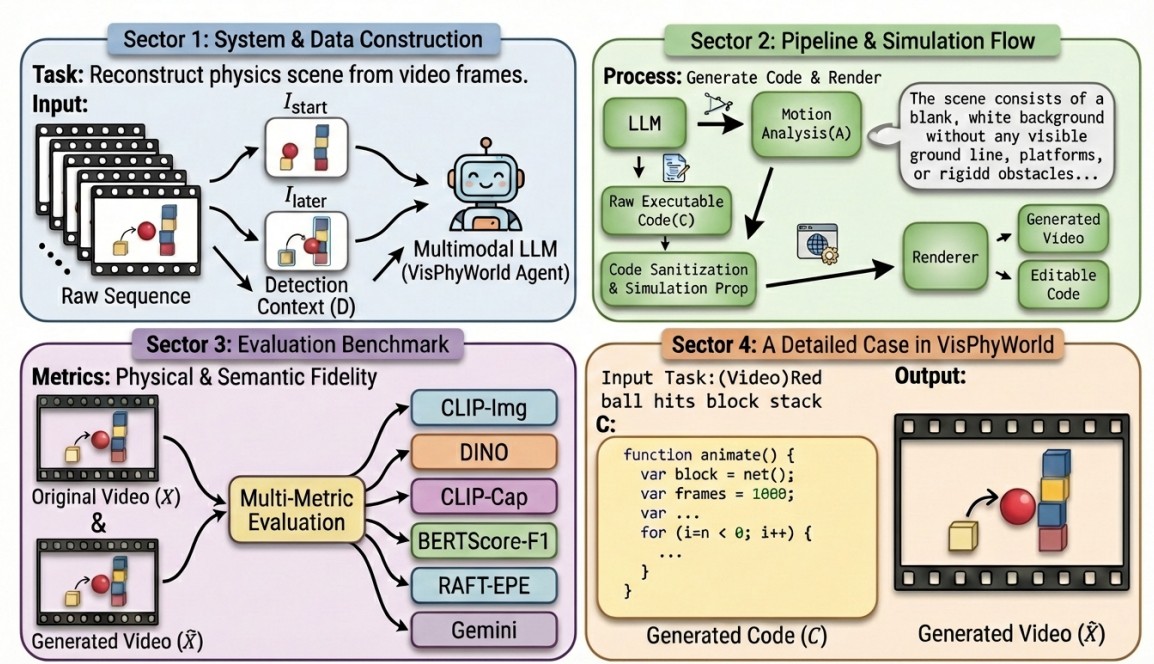

*Figure 3.* **VisPhyWorld Framework. (1) System & Data Construction:** We process raw video sequences to extract key frames ($I_{start}$, $I_{later}$) and detection contexts using multimodal agents. **(2) Pipeline & Simulation Flow:** An LLM-based agent performs motion analysis and generates raw executable code, which is then sanitized and rendered. **(3) Evaluation Benchmark:** We propose a multi-metric benchmark integrating semantic and physical fidelity to compare generated videos $\hat{X}$ with ground truth $X$. **(4) A Detailed Case:** A example illustrating how VisPhyWorld translates a collision scene (red ball hits block stack) into executable simulation logic.

cally plausible outcomes from visual observations. This has been studied through video prediction benchmarks that evaluate the consistency of predicted future dynamics, as well as Violation-of-Expectation (VoE) paradigms (Riochet et al., 2020; Margoni et al., 2024; Jassim et al., 2024), which assess whether physically implausible events elicit higher predictive surprise. These approaches are well suited to generative world models with explicit prediction objectives. However, they do not naturally extend to MLLMs, which primarily produce textual outputs rather than predictive distributions and therefore cannot be evaluated using likelihood-based or generative video protocols (Garrido et al., 2025). Efforts on several prominent datasets and benchmarks have been made (Rajani et al., 2020; Yi et al., 2020; Baradel et al., 2020), including Phyre (Bakhtin et al., 2019; Li et al., 2024), Physion (Bear et al., 2022; Tung et al., 2023), and IntPhys (Riochet et al., 2020; Bordes et al., 2025), have been proposed to evaluate intuitive physics using videos generated from physics engines. More recent benchmarks such as PhysicsIQ (Motamed et al., 2025b), PhyGenBench (Meng et al., 2024), and WorldModelBench (Li et al., 2025) extend this setting to generative video models, focusing on whether predicted videos exhibit physically plausible and temporally consistent dynamics. In parallel, researchers have developed MLLM–based evaluators (Motamed et al., 2025a), such as VideoPhy (Bansal et al., 2024; 2025) and VideoScore (He

et al., 2024b; 2025), to assess physical understanding in multimodal models. These approaches typically formulate evaluation as recognition-based tasks like VQA. While effective for probing high-level physical knowledge, such protocols make it difficult to determine whether model performance reflects genuine physical reasoning or reliance on appearance-based heuristics and dataset-specific biases. Our framework complements previous works by requiring explicit and executable physical hypotheses evaluated through simulation. Table 1 compares our work with prior works.

**Executable World Representations for Visual and Motion Generation.** Representing visual scenes as executable programs is a foundational paradigm in computer graphics and simulation, where structured code specifies objects, motion, and physical interactions to enable interpretable and controllable world representations (Foley et al., 1996). Recent advances in multimodal large language models have begun to enable the generation of executable code for visual content. Early efforts primarily focus on static visualizations, such as data plots and vector graphics, translating high-level semantic intent into low-level graphical instructions (Galimzyanov et al., 2024; Yang et al., 2024; Goswami et al., 2025; Ni et al., 2025b;a; Rodriguez et al., 2024; Yang et al., 2025; Lin et al., 2025). These methods demonstrate the feasibility of using code as a structured intermediate between language and visual output. Subsequent work extends

code-based generation to animations and motion, enabling programmatic specification of object trajectories and temporal behaviors (Zhang et al., 2023; He et al., 2024a; Liu et al., 2024; Lv et al., 2024; Ku et al., 2025). While these approaches show that MLLM can generate executable programs that produce coherent motion, they are primarily designed for content creation or presentation, and rarely assess whether the generated programs correspond to physically consistent dynamics or reflect an underlying understanding of physical laws. In contrast to prior work that treats executable visual generation as an end goal, our work uses executable world representations as a diagnostic interface for physical reasoning. Rather than evaluating visual realism or animation quality, we assess whether models can reconstruct and resimulate physically consistent dynamics from visual observations to enable direct inspection.

## 3. VisPhyWorld

We introduce VisPhyWorld, a framework that uses MLLM to interpret visual observations and reconstruct the underlying physical scene as executable code. We evaluate the rendered outputs under a unified protocol using a multi-metric suite.

### 3.1. Problem Definition

We focus on 2D and 3D physical scenes involving common interactions, e.g., ball collisions and box sliding. We represent each scene as a sequence of image frames $I$ with three color channels as in Equation 1, where $H$, $W$, and $T$ denote frame height, frame width, and number of frames, respectively.

$$X = (I_t)_{t=1}^T, \qquad I_t \in \mathbb{R}^{3 \times H \times W}, \qquad (1)$$

**Input.** Given a scene, the MLLM backbone receives $\{I^{\text{start}}, I^{\text{later}}, D\}$. We select two key frames from $X$, where $I^{\text{start}} = I_{t_s}$ and $I^{\text{later}} = I_{t_l}$, typically corresponding to an early frame and a later frame (e.g., $t_s = 1$, $t_l = 10$). Optionally, we provide a detection context $D$ for $I^{\text{start}}$ listing objects with categories, bounding boxes, and coarse attributes (details in Appendix B.2). We obtain $D$ with GPT-5.2 (OpenAI, 2025c) on the first frame and parsing its output into a structured object list; if unavailable, we set $D = \varnothing$.

**Outputs.** VisPhyWorld produces four interpretable artifacts: (i) a textual motion analysis $A \in \mathcal{Y}^{\text{text}}$; (ii) a machine-readable first-frame JSON specification $S$ encoding object layout and inferred parameters; (iii) an executable program $C \in \mathcal{Y}^{\text{code}}$; and (iv) a rendered video $\hat{X} = (\hat{I}_t)_{t=1}^{\hat{T}}$ obtained by executing the executable program $C$.

### 3.2. VisPhyWorld Architecture

$$(I^{\text{start}}, I^{\text{later}}, D) \xrightarrow{f_{\text{LLM}}} (A, S, C) \xrightarrow{R_{\text{phys}}} \hat{X}. \qquad (2)$$

VisPhyWorld implements a composite mapping as stated in Equation 2. We include $A$ as a lightweight, text-only diag-

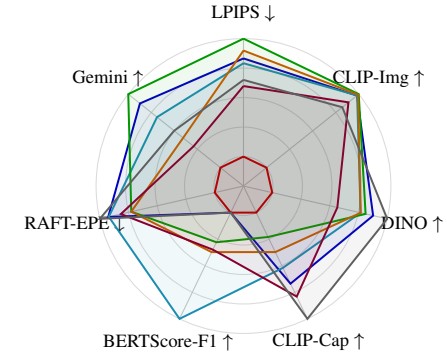

| Metric | GPT-5 | GPT-4.1 | Gemini 3-Pro | Claude 4.5 | Qwen3 VL | SVD | Veo-3.1 |
|---|---|---|---|---|---|---|---|
| LPIPS ↓ | 0.1736 | 0.1818 | 0.1399 | 0.1602 | 0.2207 | 0.3408 | 0.2102 |
| CLIP-Img ↑ | 0.8930 | 0.8933 | 0.8973 | 0.8957 | 0.8717 | 0.6677 | 0.8564 |
| DINO ↑ | 0.8556 | 0.8304 | 0.8405 | 0.8305 | 0.7837 | 0.6528 | 0.8839 |
| CLIP-Cap ↑ | 0.2632 | 0.2610 | 0.2567 | 0.2588 | 0.2650 | 0.2533 | 0.2681 |
| BERTScore-F1 ↑ | 0.8436 | 0.8522 | 0.8460 | 0.8468 | 0.8466 | N/A | N/A |
| RAFT-EPE ↓ | 33.65 | 33.71 | 36.20 | 36.20 | 35.05 | 45.46 | 32.71 |
| Gemini ↑ | 3.50 | 3.06 | 3.80 | 2.39 | 2.12 | 1.43 | 2.62 |

*Figure 4.* **Key metrics on VisPhyBench.** We compare code-driven reconstruction (multiple MLLMs) against pixel-space baselines (Veo 3.1 and SVD) under the unified evaluation protocol.

nostic of the model's basic scene understanding: whether it can correctly describe salient motions and interactions between the key frames, separately from code generation. We treat $C$ as an explicit, falsifiable physical hypothesis: executing it with a renderer $R_{\text{phys}}$ under a fixed configuration yields $\hat{X}$, separating hypothesis construction from execution and enabling controlled comparisons across LLM backbones. To ensure a well-defined evaluation, we apply lightweight validation prior to execution and allow a single automatic repair attempt upon failure; if both attempts fail, we fall back to a minimal valid scene. Further implementation details, including prompt templates and renderer settings, are deferred to Appendix B, with robustness analyses in Appendix B.5.

### 3.3. Benchmark, Metrics, and Baselines

**Dataset Construction.** We build on and extend the 2D data from the PhyWorld dataset (Kang et al., 2025), using the PHYRE engine (Bakhtin et al., 2019) for rendering to form the 2D subset of VisPhyBench. We additionally curate a 3D subset rendered with Three.js and simulated using Cannon.js for rigid-body dynamics. Overall, VisPhyBench comprises 108 templates and 209 videos, each paired with first-frame JSON annotations. VisPhyBench scenes are annotated with coarse difficulty levels, as summarized in Table 3. We construct a small test split by subsampling from the full dataset to enable rapid sanity checks and lightweight evaluation. Eight STEM-trained annotators rate each raw clip on a 1–5

*Table 2.* **Overall leaderboard on VisPhyBench**. Columns are grouped into: reconstruction–perceptual quality (LPIPS), visual semantic consistency (CLIP-Img, DINO), text–video and analysis-text consistency (CLIP-Cap, BERTScore-F1), motion / physical plausibility (RAFT-EPE), and holistic overall quality (Gemini). Higher is better (↑), lower is better (↓). "–" denotes metrics that are unavailable or not applicable for a given method.

| Model | Reconst. & Perceptual | Visual Semantic Consistency | | Text–Video & Analysis-Text | | Motion / Physical Plausibility | Holistic Quality |
|---|---|---|---|---|---|---|---|
| | LPIPS↓ | CLIP-Img↑ | DINO↑ | CLIP-Cap↑ | BERTScore-F1↑ | RAFT-EPE↓ | Gemini↑ |
| Ours (GPT-5, threejs) | 0.1736 | 0.8930 | 0.8556 | 0.2632 | 0.8436 | 33.6473 | 3.50 |
| Ours (GPT-5, p5js) | 0.2926 | 0.8134 | 0.7580 | 0.2331 | 0.8360 | 34.3433 | 3.52 |
| Ours (GPT-4.1, threejs) | 0.1818 | 0.8933 | 0.8304 | 0.2610 | **0.8522** | 33.7110 | 3.06 |
| Ours (GPT-4.1, p5js) | 0.3520 | 0.7545 | 0.6786 | 0.2192 | 0.8253 | 37.6993 | 2.15 |
| Ours (Gemini-3-Pro, threejs) | **0.1399** | **0.8973** | 0.8405 | 0.2567 | 0.8460 | 36.2030 | **3.80** |
| Ours (Gemini-3-Pro, p5js) | 0.3302 | 0.7460 | 0.6721 | 0.2184 | 0.8396 | 33.1013 | 2.35 |
| Ours (Claude Sonnet 4.5, threejs) | 0.1602 | 0.8957 | 0.8305 | 0.2588 | 0.8468 | 36.1985 | 2.39 |
| Ours (Claude Sonnet 4.5, p5js) | 0.3250 | 0.7612 | 0.7098 | 0.2177 | 0.8224 | 34.1425 | 2.56 |
| Ours (Qwen3-VL-Plus, threejs) | 0.2207 | 0.8717 | 0.7837 | **0.2650** | 0.8466 | 35.0493 | 2.12 |
| Ours (Qwen3-VL-Plus, p5js) | 0.5478 | 0.6446 | 0.5478 | 0.2032 | 0.8358 | **20.8187** | 1.46 |
| SVD (img2vid) | 0.3408 | 0.6677 | 0.6528 | 0.2533 | – | 45.4606 | 1.43 |
| Veo-3.1 | 0.2102 | 0.8564 | **0.8839** | **0.2681** | – | 32.7145 | 2.62 |

scale (higher indicates greater difficulty), and we use the mean rating as the final difficulty score. The mean score is then mapped to easy, medium, or hard using fixed, interpretable cutoffs aligned with the rating scale (easy = 1–2, medium = 3, hard = 4–5). The resulting distribution is naturally skewed, reflecting the relative rarity of challenging interactions in our template set. Scenes cover diverse object configurations (stacks, ramps, collisions) and motion patterns (slides, bounces, topples). For the 2D subset, the camera is fixed and orthographic; for the 3D subset, we use a fixed perspective camera. In both settings, the background is set to white to focus on physical dynamics. Since templates are executable programs instantiated by sampling seeds, we summarize template composition and object statistics in Appendix B.3. Inputs ($I^{start}$, $I^{later}$, $D$) follow Section 3.1.

*Table 3.* Difficulty stratification of VisPhyBench splits

| Split | Easy | Medium | Hard |
|---|---|---|---|
| sub (209) | 114 | 67 | 28 |
| test (49) | 29 | 17 | 3 |

**Evaluation Metrics.** We report per-metric means over all scenes and group metrics into five families. **(1)** Reconstruction and perceptual quality. We report PSNR (Huynh-Thu & Ghanbari, 2008) and SSIM (Wang et al., 2004) for frame-wise reconstruction, together with LPIPS (Zhang et al., 2018), FSIM (Zhang et al., 2011), VSI (Zhang et al., 2014), and DISTS (Ding et al., 2020) to compute on aligned frame pairs. **(2)** Visual semantic consistency. We compute CLIP-based image similarity (CLIP-Img) (Radford et al., 2021) and DINO feature similarity (Caron et al., 2021), which emphasize object identity and scene layout beyond exact pixels. **(3)** Text–video and analysis-text consistency. We compute CLIP text–image similarity (CLIP–Cap) (Radford et al., 2021) between the analysis and sampled video frames, and use ROUGE-L (Lin, 2004) and BERTScore-F1 (Zhang et al., 2020) to compare the analysis

with a GPT-generated reference description derived from the ground-truth video. **(4)** Motion and physical plausibility. We use RAFT-based optical-flow diagnostics (Teed & Deng, 2020) with automatic temporal alignment, reporting RAFT end-point error (EPE) and the estimated temporal offset (and, when relevant, flow magnitude and angular statistics) to quantify motion consistency. Because flow discrepancy alone can be misleading as a proxy for physical plausibility, we interpret RAFT metrics jointly with holistic perceptual-/physics judgments rather than using RAFT-EPE in isolation; as discussed in Section 4. **(5)** Subjective overall quality. We use a Gemini-2.5-Pro video–video judge (1–10) with a short textual justification, and separately report pipeline success rate based on whether a valid video is produced.

**Video Model Baselines.** We include Stable Video Diffusion (SVD) img2vid (Blattmann et al., 2023), conditioned only on $I^{start}$, and Veo-3.1, conditioned on ($I^{start}$, $I^{later}$, prompt) in pixel space.

### 3.4. Engine Evaluation and Selection

We evaluate four rendering backends, i.e., Three.js (three.js contributors, 2026), P5.js (p5.js contributors, 2026), SVG (Scalable Vector Graphics), and Manim (The Manim Community Developers, 2024), to understand how the choice of visualization engine affects multimodal LLM-based reconstruction. As shown in Figure 1, a consistent pattern emerges: Three.js and P5.js achieve markedly better reconstruction and motion fidelity because they support native integration with rigid-body physics solvers, allowing the generated programs to offload gravity, contact constraints, friction, and collision response to a physically grounded engine. In contrast, SVG and Manim are primarily non-physics-based rendering systems: they excel at deterministic drawing and scripted animation, but lack intrinsic rigid-body dynamics. In our experimental setting, SVG and Manim serve as non-interactive, script-based backends and do not expose a comparable physics API or closed-loop simulation stepping; consequently, as illustrated in Figure 1, they

*Table 4.* Average PSNR/SSIM and generation success rate.

| Model | Engine | PSNR ↑ | SSIM ↑ | Success ↑ |
|-------|--------|--------|--------|-----------|
| GPT-5 | Three.js | 20.54 | 0.9370 | 0.990 |
| GPT-5 | p5.js | 16.36 | 0.7440 | 0.979 |
| GPT-4.1 | Three.js | 19.74 | 0.9337 | 0.948 |
| GPT-4.1 | p5.js | 14.83 | 0.6830 | **1.000** |
| Gemini-3-Pro | Three.js | **21.26** | **0.9445** | 0.957 |
| Gemini-3-Pro | p5.js | 15.57 | 0.6943 | 0.963 |
| Claude Sonnet 4.5 | Three.js | 20.75 | 0.9406 | 0.995 |
| Claude Sonnet 4.5 | p5.js | 15.36 | 0.7160 | **1.000** |
| Qwen3-VL-Plus | Three.js | 18.66 | 0.9306 | 0.936 |
| Qwen3-VL-Plus | p5.js | 9.14 | 0.4296 | **1.000** |
| SVD (img2vid) | – | | 14.44 | 0.8802 | **1.000** |
| Veo-3.1 | – | | 20.04 | 0.9354 | **1.000** |

often yield physically implausible behaviors, such as objects remaining static or interpenetrating. Importantly, this gap suggests a limitation of current MLLM: without access to a true physics solver, they fail to consistently infer and apply Newtonian dynamics from visual evidence, and instead revert to heuristic motion scripting. For this work, we therefore prioritize Three.js and P5.js so that our evaluation emphasizes physically grounded re-simulation rather than non-physical animation artifacts.

## 4. Experiments

**Evaluation setup.** We evaluate VisPhyWorld and all baselines on VisPhyBench. For each configuration, we generate one video per scene, compute all metrics, and report per-metric means over the evaluation split; unless otherwise stated, higher is better. We consider five multimodal LLM backbones: GPT-5 (OpenAI, 2025b), GPT-4.1 (OpenAI, 2025a), Gemini-3-Pro (Google AI for Developers, 2026), Claude Sonnet 4.5 (Anthropic, 2025), and Qwen3-VL-Plus (Alibaba Cloud, 2026). We evaluate two code backends, Three.js (three.js contributors, 2026) and P5.js (p5.js contributors, 2026). All LLM runs use the same prompt and two key frames; only the model and engine identifiers change. For each run, we aggregate metrics into five families: reconstruction & perceptual quality, visual semantic consistency, text–video & analysis-text consistency, motion (automatic RAFT-based metrics (Teed & Deng, 2020)), and subjective overall quality (Gemini-2.5-Pro (Google, 2025) judge). We observe consistent trends across scenes; per-scene metric distributions and significance analyses are reported in Appendix D.2, Fig. 21.

**Overall leaderboard.** Table 2 summarizes performance across five metric families on VisPhyBench, while Table 4 reports pixel-space fidelity (PSNR, SSIM) and execution success rates. Overall, most models achieve strong reconstruction and perceptual scores and maintain reasonable visual-semantic consistency; these results support our central claim that, once the task is cast as executable hypotheses under a fixed physics engine, most modern MLLM can re-

construct synthetic physical events with high fidelity, and the remaining gaps become diagnosable rather than opaque. First, our benchmark jointly evaluates visual reconstruction/semantics and language-mediated reasoning, and we observe that these two dimensions can dissociate across models. Some backends achieve the strongest perceptual and semantic alignment to the reference frames, with low LPIPS and high CLIP and DINO, indicating that they are effective at extracting correct object identities and global layouts from the visual input. For example, Gemini-3-Pro (threejs) attains the lowest LPIPS together with the highest CLIP-Img, and it also yields the strongest pixel-level reconstruction in Table 4. In contrast, Veo-3.1 does not produce an executable simulator and thus lacks interpretable intermediate states for diagnosis. Others attain the best analysis-text agreement, suggesting stronger descriptive and causal narration even when perceptual scores are not the top: GPT-4.1 (threejs) achieves the highest BERTScore-F1 despite a higher LPIPS than Gemini-3-Pro (threejs). This dissociation implies that the benchmark is not merely measuring overall model quality; instead, it teases apart seeing the scene from explaining it.

Second, the choice of code backend affects reconstruction quality. Across LLMs, Three.js variants yield better perceptual reconstruction than their P5.js counterparts, as reflected by lower LPIPS in most pairs, despite sharing the same conditioning inputs and prompt. Concretely, for GPT-5, switching to Three.js reduces LPIPS error by nearly 40% ($0.29 \rightarrow 0.17$) and boosts structural similarity (SSIM) from 0.74 to 0.94. Visually, this corresponds to better preservation of object identity, as illustrated in Figure 1. Since the physics engine is fixed, this performance gap confirms that the simulator's expressivity affects the model's ability to ground visual evidence. In other words, program structure and simulator interface materially affect how well a model can translate visual evidence into a stable physical hypothesis.

Third, pixel-space baselines exhibit a complementary profile: they can score competitively on some frame-feature semantics, but their failures are harder to attribute to specific physical causes, such as friction, restitution, or contact timing, since the generation process does not expose interpretable latent variables. Veo-3.1 attains reasonable semantic similarity, for example reaching DINO $\sim 0.88$, yet it does not expose an explicit simulator state for diagnosis or controlled interventions and often exhibits deficiencies in physical understanding by producing trajectories with implausible motion or contact events(see Sec. 4.2). Conversely, our code-driven approach maintains competitive semantic and motion scores while exposing executable states; e.g., GPT-5 (threejs) achieves DINO 0.8556 with RAFT-EPE 33.6473. This enables controlled interventions (e.g., varying friction/mass while holding layout fixed) that can isolate whether an error originates from object discovery, state ini-

*Table 5.* Text analysis consistency on the VisPhyBench. We compare model analyses against GPT-5.1-generated reference descriptions using ROUGE-L and BERTScore.

| Backbone | Tool | ROUGE-L F1 ↑ | BERTScore F1 ↑ |
|---|---|---|---|
| GPT-5 | Three.js | 0.2186 | 0.8436 |
| GPT-5 | p5.js | 0.2057 | 0.8360 |
| GPT-4.1 | Three.js | **0.2383** | **0.8522** |
| GPT-4.1 | p5.js | 0.1689 | 0.8253 |
| Gemini-3-Pro | Three.js | 0.2141 | 0.8460 |
| Gemini-3-Pro | p5.js | 0.1886 | 0.8396 |
| Claude Sonnet 4.5 | Three.js | 0.2168 | 0.8468 |
| Claude Sonnet 4.5 | p5.js | 0.1599 | 0.8224 |
| Qwen3-VL-Plus | Three.js | 0.2022 | 0.8466 |
| Qwen3-VL-Plus | p5.js | 0.1733 | 0.8358 |

tialization, or contact modeling, aligning with our goal of turning "physics understanding" into a testable, executable object.

Finally, we report holistic quality using a Gemini-2.5-Pro judge (see Appendix C.6), which aggregates multiple perceptual and physical cues into a single preference signal. This holistic score aligns with strong visual alignment for some backends. For example, Gemini-3-Pro (threejs) reaches the highest Gemini score, 3.80, while penalizing visibly implausible or degraded generations. For instance, Qwen3-VL-Plus (p5js) scores 1.46 alongside poor perceptual/semantic alignment. This judge complements the automatic metrics by capturing visually salient failure modes (e.g., missing motion, implausible contacts) that may not be fully reflected in any single metric family. Together, these results indicate that VisPhyBench and our metric suite jointly provide a multi-view, diagnostic measurement of LLM visual and physical competence under executable simulation.

**Motion and physical plausibility.** Assessing physical plausibility requires balancing raw motion statistics with perceptual coherence. While RAFT-EPE measures optical flow discrepancy, relying on it in isolation can be misleading; for instance, Qwen3-VL-Plus in P5.js attains the lowest RAFT-EPE, 20.82, despite poor reconstruction fidelity in Table 4. Consequently, we adopt a joint evaluation strategy: a model is considered to demonstrate valid physical understanding only when it achieves favorable RAFT-EPE, which reflects motion-trajectory agreement as detailed in Appendix C, and a high Gemini holistic score, which reflects perceptually coherent outcomes under a physics-focused rubric in Appendix C.6. Furthermore, the Gemini evaluator returns a textual justification that explicitly comments on physical plausibility, including collisions, contact consistency, and implausible motion, providing a qualitative sanity check alongside the quantitative flow metrics.

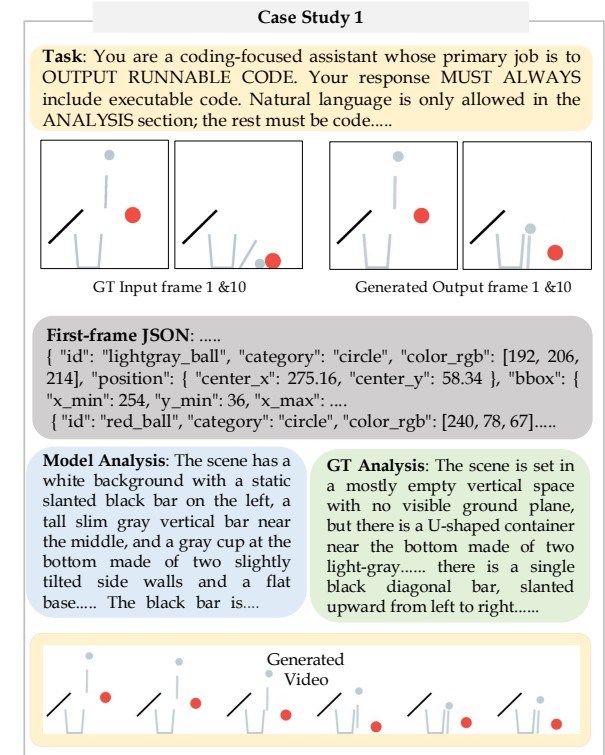

*Figure 5.* This case shows that VisPhyWorld exhibits strong physical grounding, correctly simulating the collision dynamics. More examples are in the Appendix.

*Table 6.* Ablation on iterative self-repair (retry) on the VisPhyBench. "No retry" counts only samples that succeed on the first generation+render attempt; "1 retry" allows one additional attempt with renderer error feedback appended to the prompt.

| Engine | Success (no retry)↑ | Success (1 retry)↑ | Fixed by retry↑ |
|---|---|---|---|
| Three.js | 0.979 | 0.990 | 0.010 |
| P5.js | 0.853 | 0.979 | 0.126 |

## 4.1. Ablation on iterative self-repair (retry)

VisPhyWorld includes an iterative self-repair mechanism: if the first generation+render attempt fails, we append a concise renderer error log tail and the previous attempt to the next LLM call and retry once. Table 6 reports the success rate on the VisPhyBench with and without this retry mechanism. Overall, the retry mechanism provides substantial gains, suggesting that most failures are due to correctable surface-level issues (e.g., missing canvas hooks, minor API misuse, or initialization errors) rather than irrecoverable scene-understanding errors.

## 4.2. Case Study

We present a diagnostic case study, shown in Figs. 5 and 6, featuring gravity-driven multi-body interactions that require precise physical reasoning. GPT-5 in Three.js shows strong physical grounding by correctly simulating the collision

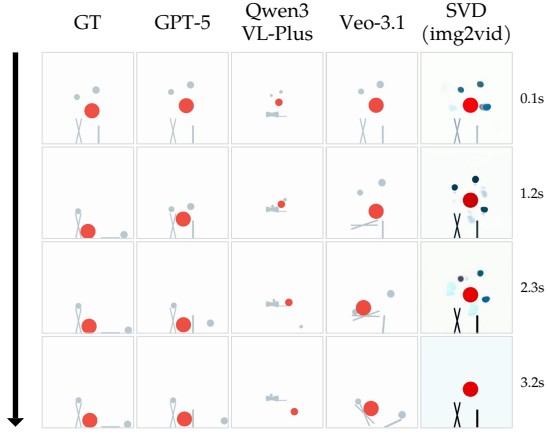

Figure 6. GPT-5 reconstructs object identities and collision dynamics most faithfully over time. Pixel-space baselines (Veo-3.1 and SVD/img2vid) generate trajectories with implausible motion/contact events due to the lack of an explicit physics hypothesis.

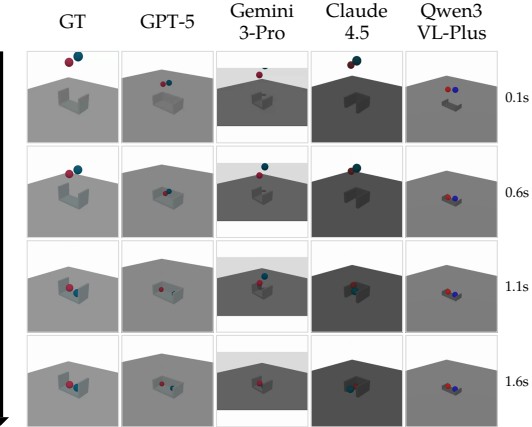

Figure 7. This example highlights the dissociation between semantic alignment and correct physical dynamics: although Claude shows clear reconstruction deficits, its visual-semantic scores remain relatively high.

dynamics, achieving Gemini 10.0 with DINO 0.926. In contrast, pixel-space baselines such as Veo-3.1 achieve high semantic similarity, reaching DINO 0.835, but fail on event logic with Gemini 2.0, indicating plausible appearance with hallucinated dynamics. The case also motivates joint evaluation: Qwen3-VL-Plus attains low RAFT-EPE, 121.30 versus 118.66 for GPT-5, by producing static or empty outputs, yet is penalized by Gemini with a score of 4.0. Together, these results show that optical-flow errors alone are insufficient; credible physical understanding requires both correct motion and holistic visual coherence.

Figure 7 extends our diagnostic analysis beyond 2D templates to a perspective-rendered 3D scene with depth-dependent contacts and occlusions. Consistent with our 2D findings, we observe the same conclusion in 3D: strong appearance matching alone does not guarantee physically faithful dynamics. Valid physical understanding is only evidenced when both motion dynamics and holistic visual coherence are simultaneously satisfied. For example, Claude-4.5 and Qwen3-VL-Plus exhibit clear reconstruction deviations in this sample, yet their visual-semantic scores do not separate substantially from other models, highlighting a dissociation between semantic alignment and correct physical dynamics. More broadly, the 3D setting is noticeably more challenging for current MLLMs, underscoring the necessity of incorporating 3D scenes when evaluating reconstruction-based physical reasoning.

## Limitations and Discussion

While VisPhyWorld shows promising results on physics-aware video generation and evaluation, it has several limitations. First, our experiments are conducted on synthetic, simulator-driven scenes with controlled object layouts and camera motion, so generalization to high-resolution, in-the-wild videos remains untested. Fundamentally limited by

the capabilities of current MLLM and the complexity of modern engines, VisPhyWorld can reliably generate code only for relatively simple rigid-body scenes: although we experimented with large engines such as Unreal Engine and Blender, we found that, without human intervention, existing MLLMs cannot, within a small fixed number of calls, autonomously produce and repair simulation code to render a stable, visually plausible video in these more complex environments. Finally, we currently target relatively short clips with moderate motion complexity, and do not explicitly address long-horizon interactions, complex 3D reasoning, or stylized or heavily cluttered scenes, which we leave for future work. Future work could integrate stronger 3D perception for scene initialization and agentic workflows with domain-specific fine-tuning.

## Conclusions

In this work, we introduced VisPhyWorld, a framework that advances the evaluation of physical understanding by requiring MLLMs to reconstruct scenes as executable code, thereby decoupling visual mimicry from physically grounded reasoning. By benchmarking state-of-the-art models on our proposed VisPhyBench, we exposed a consistent dichotomy: while current models excel at semantic scene parsing, they struggle with precise physical parameterization; indeed, when forced to commit to an executable hypothesis, models that rely on pixel-space generation often fail to reproduce even basic Newtonian dynamics. Our findings suggest that progress toward robust world modeling may benefit from moving beyond purely statistical pattern matching in pixel space toward hybrid representations that ground visual perception in verifiable, executable physical laws. We believe this direction offers a concrete path toward more transparent and verifiable evaluations of physical understanding.

## Impact Statement

This work advances the interpretability and reliability of generative AI by transforming opaque video prediction into transparent, executable code, which is essential for deploying reliable world models in safety-critical domains like robotics. By grounding generation in explicit symbolic logic, our approach offers a mechanism to audit and verify physical hallucinations, potentially mitigating risks associated with black-box simulations. While improved physical reasoning capabilities could enhance synthetic media generation, the inherent falsifiability and inspectability provided by our code-driven paradigm serve as a crucial safeguard against unverifiable generation.

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

## A. Case Study

### Case Study 2

**Task**: You are a coding-focused assistant whose primary job is to OUTPUT RUNNABLE CODE. Your response MUST ALWAYS include executable code. Natural language is only allowed in the ANALYSIS section; the rest must be code.....

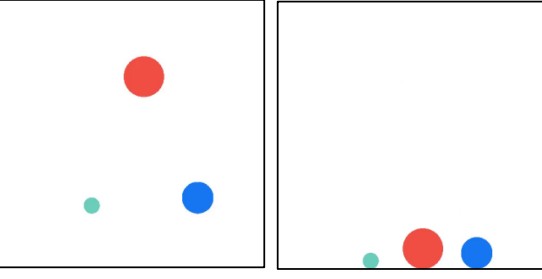 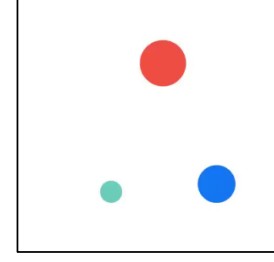 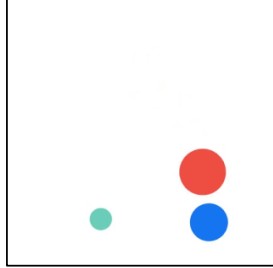

GT Input frame 1 &10 · Generated Output frame 1 &10

**First-frame JSON**: .....
{ "id": "red_ball", "category": "circle", "color_rgb":[240, 78, 67], "position": { "center_x": 280.04, "center_y": 147.27 }, "bbox": { "x_min": 240, "y_min": 102, "x_max": ....
{ "id": "blue_ball", "category": "circle", "color_rgb": [22, 118, 241], "position": { "center_x": 383.73....

**Model Analysis**: The scene is an open white area with an implied flat ground along the very bottom of the frame; there are no black fixed bars or obstacles. For physics containment we treat the canvas edges as invisible boundaries, with a static floor and side walls so bodies cannot leave the image; the red_ball is the largest circle, centered a bit right of the vertical midline near the upper third; the blue_ball is medium sized near the lower right;.....

**GT Analysis**:The scene consists of a blank, white background with no visible ground line, platforms, walls, or other static supports. There are no ramps, pegs, or obstacles; the space appears open and unobstructed in both horizontal and vertical directions. The objects seem to float in this empty 2D field, implying that any motion will be governed mainly by gravity in the vertical direction and free translation horizontally.\n\nThere are three dynamic circular objects ......

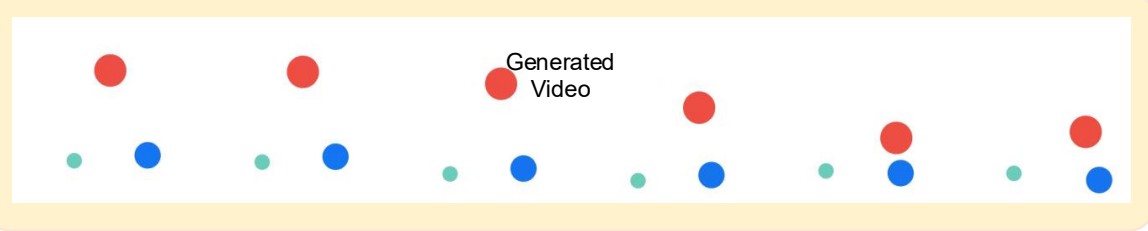

Generated Video

*Figure 8.* A detailed case study (ID 2).

## Case Study 3

**Task**: You are a coding-focused assistant whose primary job is to OUTPUT RUNNABLE CODE. Your response MUST ALWAYS include executable code. Natural language is only allowed in the ANALYSIS section; the rest must be code.....

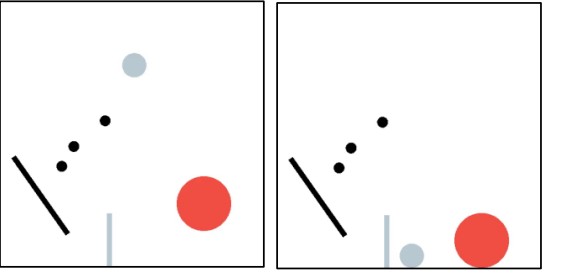
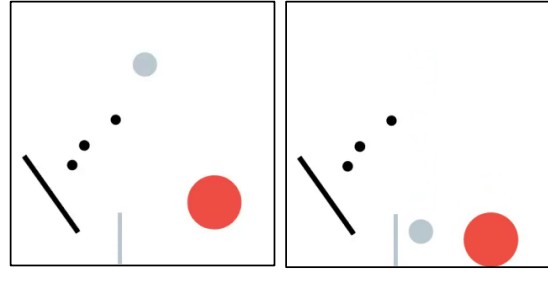

GT Input frame 1 &10         Generated Output frame 1 &10

**First-frame JSON**: .....
{ "id": "gray_ball", "category": "circle", "color_rgb": [0, 0, 0], "position": { "center_x": 260.62, "center_y": 121.6 }, "bbox": { "x_min": 212, "y_min": 196, "x_max": ....
{ "id": "black_ball_1", "category": "circle", "color_rgb": [0, 0, 0], "position": { "center_x": 203.84,....

**Model Analysis**: The scene is set within a 2D vertical plane with a gravitational field pulling downwards. Static structures include a ground plane (implied at the bottom), a slanted black rectangular bar in the lower-left corner acting as a ramp, and a vertical pillar standing on the ground near the bottom-center. Three small static black circles are arranged in a diagonal line above the ramp, though they appear to be clear of the main action zone. ......

**GT Analysis**: The scene is set in empty space with a plain white background and no visible ground or enclosing walls. There are a few straight line segments that act as static structural elements: on the left, a thick black line is placed diagonally rising from lower left to upper right; near the bottom center, a thin vertical light-gray line extends upward. ......There are four circular dynamic objects. Near the top center is a medium-sized light-gray disk......

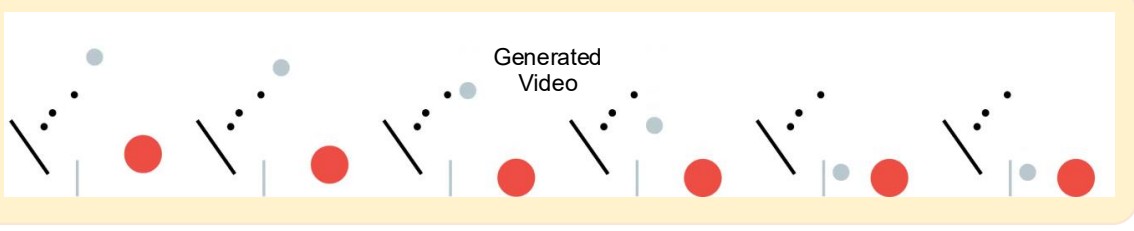

Generated Video

*Figure 9.* A detailed case study (ID 3).

## Case Study 4

**Task**: You are a coding-focused assistant whose primary job is to OUTPUT RUNNABLE CODE. Your response MUST ALWAYS include executable code. Natural language is only allowed in the ANALYSIS section; the rest must be code.....

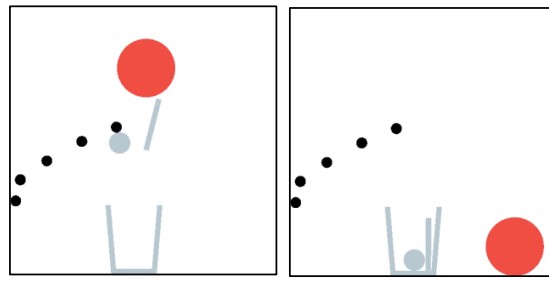
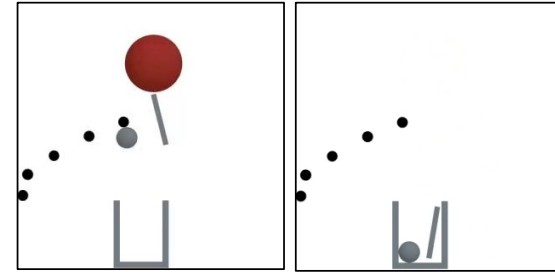

GT Input frame 1 &10                    Generated Output frame 1 &10

**First-frame JSON**: .....
{ "id": "red_ball", "category": "circle", "color_rgb": [240, 78, 67], "position": { "center_x": 261.02, "center_y": 116.06 }, "bbox": { "x_min": 288, "y_min": 256, "x_max": ....
{ "id": "line_segment", "category": "circle", "color_rgb": [184, 200, 209], "position": { "center_x":....

**Model Analysis**: The scene is set up as a 2D physics puzzle on a vertical plane. Static structures include a U-shaped gray container resting at the bottom center and a decorative arc of five small black circles on the left side, which act as fixed obstacles or markers. The ground plane is effectively the bottom edge of the image. ......

**GT Analysis**: The scene is set against a plain white background with no visible floor or ground platform. Near the bottom center there is a U-shaped static gray structure, resembling a container or goal, formed by two vertical gray bars with an open top. Near the upper central region is a large red circular object connected to a short vertical gray bar or rod that appears fixed or hinged; this rod suggests a possible obstacle or paddle-like structure attached to ......

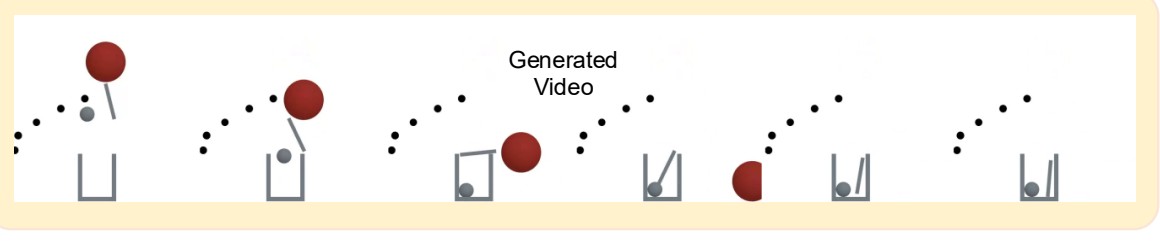

Generated Video

*Figure 10.* A detailed case study (ID 4).

## Case Study 5

**Task**: You are a coding-focused assistant whose primary job is to OUTPUT RUNNABLE CODE. Your response MUST ALWAYS include executable code. Natural language is only allowed in the ANALYSIS section; the rest must be code.....

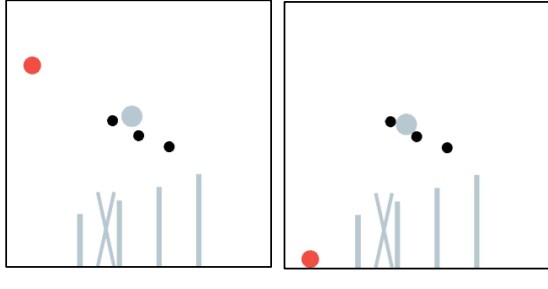 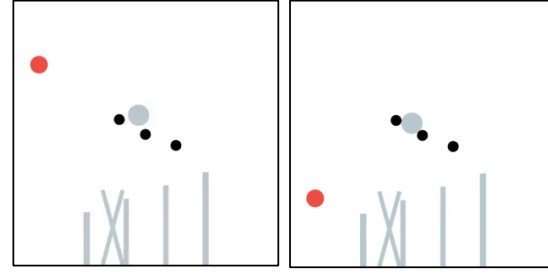

GT Input frame 1 &10                    Generated Output frame 1 &10

**First-frame JSON**: .....
{ "id": "red_ball", "category": "circle", "color_rgb": [240, 78, 68], "position": { "center_x": 48.52, "center_y": 123.48}, "bbox": { "x_min": 262, "y_min": 222, "x_max": ....
 { "id": "gray_ball", "category": "circle", "color_rgb": [0, 0, 0], "position": { "center_x": 241.7,....

**Model Analysis**: The scene presents a 2D physics setup with a mix of static structures and dynamic spheres. The static environment at the bottom consists of four gray vertical pillars of increasing height from left to right, and a tall, narrow X-shaped structure to their left. Suspended in the center are three small, fixed black circles arranged in a descending diagonal, acting as rigid pivots or obstacles ......

**GT Analysis**: The scene is set in a mostly empty vertical space with a white background and no visible ground plane. At the lower right region there are several tall, thin, vertical light-gray bars acting as fixed obstacles or posts. One bar is on the far right, another slightly left of it, and two more on the left side that cross each other in an "X" pattern, forming a loose barrier or gate. There are no ramps or horizontal platforms; these vertical elements are the primary......

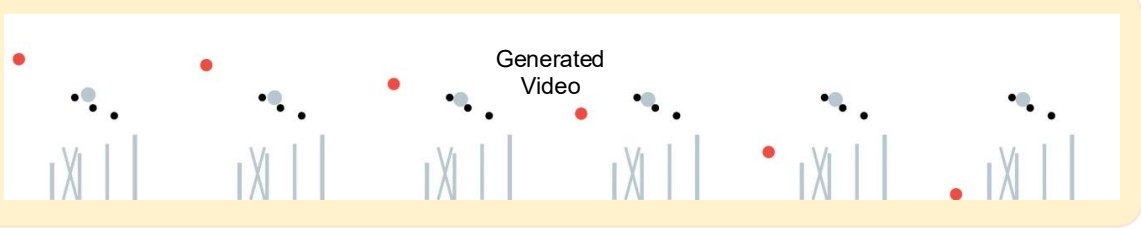

*Figure 11.* A detailed case study (ID 5).

# Case Study 6

**Task**: You are a coding-focused assistant whose primary job is to OUTPUT RUNNABLE CODE. Your response MUST ALWAYS include executable code. Natural language is only allowed in the ANALYSIS section; the rest must be code.....

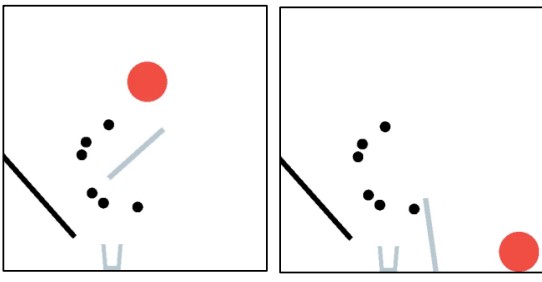 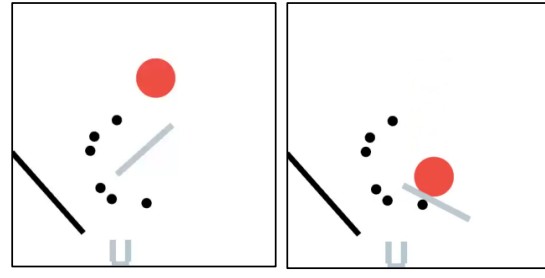

GT Input frame 1 &10                    Generated Output frame 1 &10

**First-frame JSON**: .....
{ "id": "red_ball", "category": "circle", "color_rgb": [240, 78, 68], "position": { "center_x": 279, "center_y": 146 }, "bbox": { "x_min": 312, "y_min": 202, "x_max": ....
 { "id": line_segment, "category": "circle", "color_rgb": [192, 202, 208], "position": { "center_x": 257.14,"center_y": 285.48....

**Model Analysis**: The scene is a 2D physics puzzle contained within a 512x512 area. Static structures include a thick black diagonal barrier on the left acting as a ramp, a curved cluster of six small black circular pegs in the center-left, and a U-shaped container (cup) composed of three light-blue bars at the bottom center. A ground plane is implied at the bottom edge......

**GT Analysis**: The scene appears to be a sparse environment with a white background. On the left side, there is a single thick black straight segment that is tilted upward from lower left toward upper right, acting as an inclined rigid bar or ramp. No clear ground line or enclosing boundaries are visible; the rest of the space is open. There are no obvious fixed supports other than this black bar.Dynamic objects include: (1) a large red circle located in the upper-right......

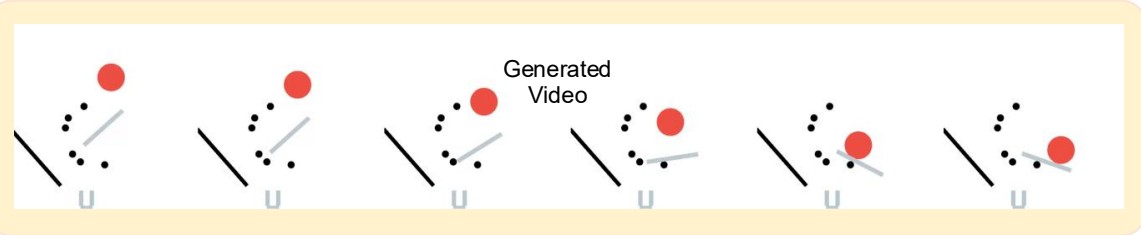

Generated Video

*Figure 12.* A detailed case study (ID 6).

## Case Study 7

**Task**: You are a coding-focused assistant whose primary job is to OUTPUT RUNNABLE CODE. Your response MUST ALWAYS include executable code. Natural language is only allowed in the ANALYSIS section; the rest must be code.....

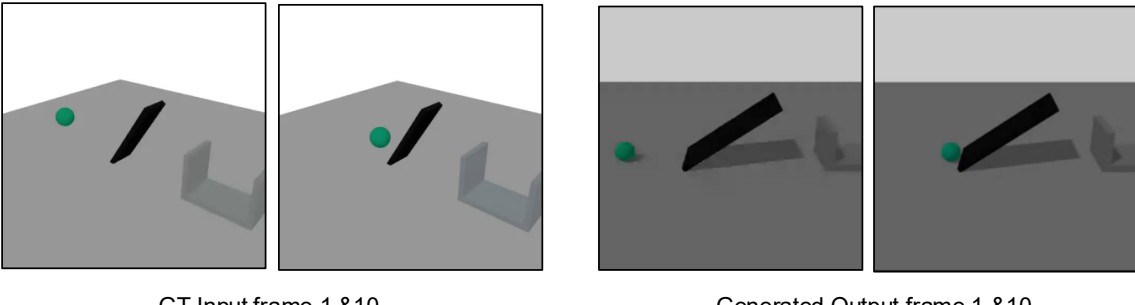

GT Input frame 1 &10                    Generated Output frame 1 &10

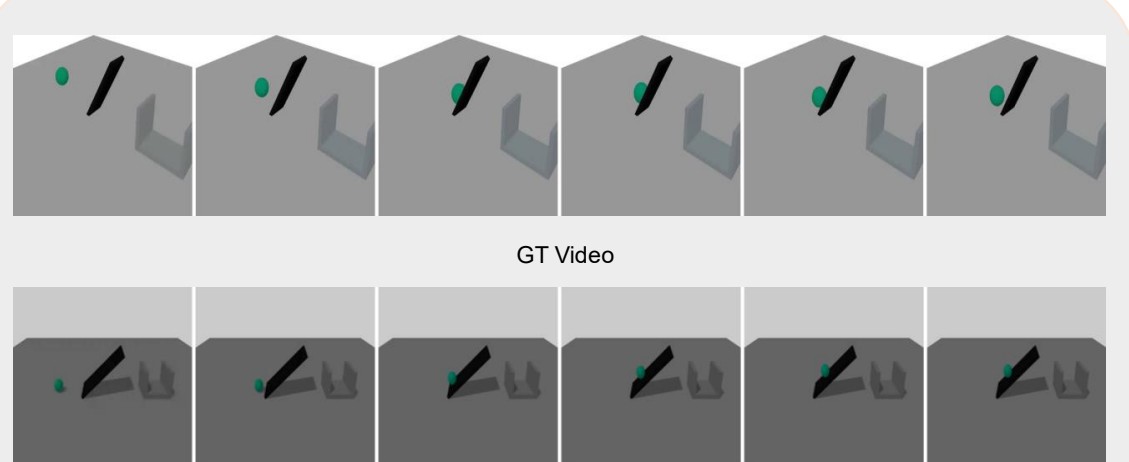

GT Video

Generated Video

**Model Analysis**: The scene is set on a large, flat, grey ground plane. In the center, a black rectangular ramp is inclined upwards from left to right, acting as a launchpad. To the right of the ramp, there is a static grey structure shaped like a U-channel or open container, consisting of a bottom slab and two vertical walls (one closer to the ramp, one further away)......

**GT Analysis**: The scene is set on a flat, light gray rectangular platform that serves as the ground plane. Near the right side of the platform there is a U-shaped......Closer to the center of the platform lies a long, thin, black rectangular plank or wedge resting on the surface and oriented diagonally, with its left end closer to the viewer and its right end pointing slightly away......There is one clearly dynamic object: a small green sphere positioned ......

*Figure 13.* A detailed case study (ID 7).

## Case Study 8

**Task**: You are a coding-focused assistant whose primary job is to OUTPUT RUNNABLE CODE. Your response MUST ALWAYS include executable code. Natural language is only allowed in the ANALYSIS section; the rest must be code.....

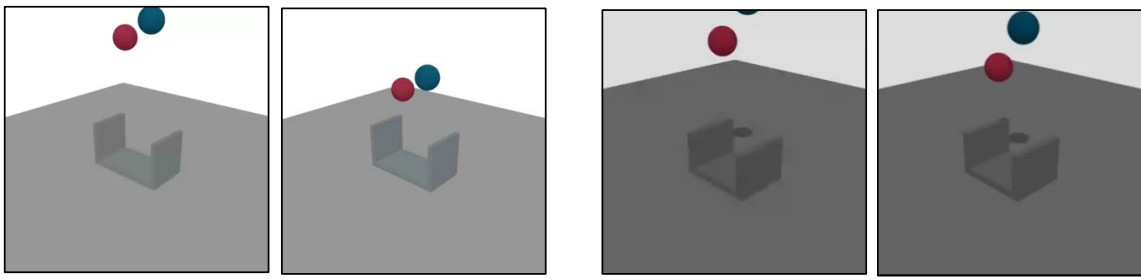

GT Input frame 1 &10       Generated Output frame 1 &10

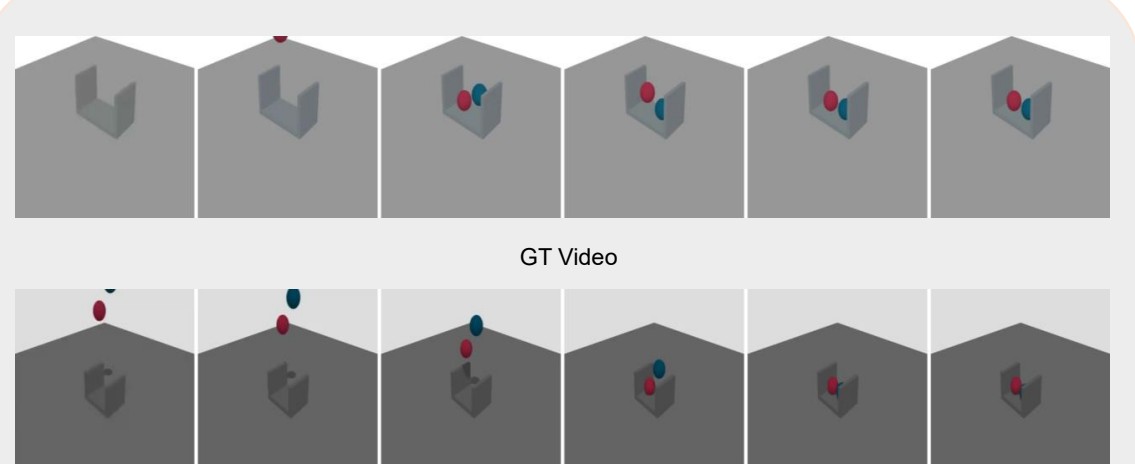

GT Video

Generated Video

**Model Analysis**: The scene consists of a static grey environment featuring a large ground plane and a U-shaped container (trough) resting on it. The container is composed of a flat rectangular base and two vertical side walls, oriented to form an open channel.Two dynamic spheres are suspended in mid-air above the container. The first is a reddish-pink sphere positioned lower......

**GT Analysis**: The scene consists of a large, flat, gray rectangular ground plane occupying most of the view. Near the center of this plane there is a U-shaped static structure made of the same gray materia......There are two dynamic objects, both spheres. One sphere is red and the other is blue...... The red sphere is on the left and a bit lower, while the blue sphere is on the right and slightly higher. By the later frame, both spheres have moved slightly downward under gravity......

*Figure 14.* A detailed case study (ID 8).

## B. Reproducibility Details

This appendix documents the reproducibility-critical components of VisPhyWorld: (i) the prompting protocol used to elicit an executable scene hypothesis, (ii) the optional detection context format, (iii) deterministic execution constraints for rendering, and (iv) robustness protocols that ensure a well-defined evaluation.

### B.1. Prompting Protocol for Scene Hypotheses

VisPhyWorld uses a single-call prompting protocol that asks the model to (1) summarize the observed motion between two keyframes and (2) propose an executable scene hypothesis that reproduces the event. To ensure comparability across models, we enforce a fixed output format and a small set of execution constraints (e.g., a single canvas and bounded duration), which are handled by the renderer (Appendix B.4). The full prompt template is shown in Figure 15.

---

**Scene Analysis & Code Generation Prompt**

You are an expert in 2D physics, rigid-body simulation, and JavaScript.
Given two key frames from a short video and an optional list of detected objects, your goals are:
**(1) Scene and motion analysis** (3–8 sentences):

- Describe the main objects (shapes, colors, approximate sizes) visible in the first frame.

- Describe how these objects move between the first and the second frame (who moves, who stays still, collisions, stacks that topple, etc.).

- Explain the likely physical causes of the motion (gravity, contact forces, friction, impulses).

**(2) Simulation code generation:** Produce ONE complete HTML document that:

- Imports the required rendering/physics libraries (or uses provided local copies if available).

- Creates a 2D-like scene with an orthographic camera.

- Adds rigid bodies (balls, boxes, planks) matching the layout of the first frame.

- Initializes positions and orientations so that the first rendered frame closely matches the first image.

- Assigns velocities or impulses consistent with the observed motion between the two images.

- Runs a physics simulation and renders frames to a single canvas element.

- Uses the provided recording helper to export a finite-duration clip.

Return your answer in the following format:
**(A) Analysis section:** plain English paragraphs.
**(B) Code section:** a single fenced block
    ```html
<!DOCTYPE html>...</html>
    ```
Do not include any other Markdown fences or extra HTML documents.

---

*Figure 15.* Full multimodal LLM prompt template used by VisPhyWorld for both motion analysis and code generation.

### B.2. Detection Context $D$

To reduce ambiguity in object discovery and initialization, VisPhyWorld can optionally provide a structured detection context $D$ for the first keyframe $I^{\text{start}}$. $D$ is a per-sample JSON annotation containing a list of objects with coarse geometry and appearance attributes. All coordinates are in pixel space with origin at the image top-left ($x$ increases rightward, $y$ increases downward).
**Schema.** Each detected object provides: (i) a unique identifier id; (ii) a coarse category (e.g., circle, rectangle, line, u_shape); (iii) an RGB color triplet color_rgb; (iv) a tight bounding box bbox as $\{$x_min, y_min,

**3D Scene Analysis & Code Generation Prompt (dataset_3D)**

You are an expert in 3D rigid-body physics, Three.js, and JavaScript.

Given two key frames from a short video (rendered from a fixed camera) and an optional list of detected objects, your goals are:

**(1) Scene and motion analysis** (3–8 sentences):

- Describe the major objects and supports visible in the first frame (shapes, colors, approximate sizes, and relative depth if evident).

- Describe how these objects move between the first and the second frame, focusing on contacts, impacts, and constraint satisfaction.

- Explain the likely physical causes of the motion (gravity, contact forces, friction, impulses).

**(2) Simulation code generation:** Produce ONE complete HTML document that:

- Uses Three.js for rendering and Cannon.js for rigid-body simulation.

- Creates a **3D** scene with a fixed *perspective* camera (no camera motion).

- Initializes objects and supports so the first rendered frame closely matches the first image.

- Assigns velocities or impulses consistent with the observed motion between the two images.

- Runs the physics simulation deterministically and renders frames to a single canvas element.

- Uses the provided recording helper to export a finite-duration clip.

Return your answer in the following format:

**(A) Analysis section:** plain English paragraphs.

**(B) Code section:** a single fenced block

```html
<!DOCTYPE html>...</html>
```

Do not include any other Markdown fences or extra HTML documents.

*Figure 16.* 3D prompt variant used for dataset_3D. It mirrors Figure 15 but switches the execution target from 2D (orthographic) to 3D (fixed perspective) while keeping the output format identical.

x_max, y_max, width, height}; (v) a centroid position.center_x/center_y; (vi) a coarse size descriptor (e.g., radius_pixels for circles, length_pixels/thickness_pixels for bars); and (vii) an optional orientation.angle_deg for elongated primitives.

**Example.**

```
{
  "image_size": {"width": 512, "height": 512},
  "coordinate_system": {"origin": "top_left", "x_axis": "to_right"...},
  "objects": [
    {"id":"red_ball","category":"circle","color_rgb":[240,78,70],
     "position":{"center_x":363.6,"center_y":155.2},
     "bbox":{"x_min":348,"y_min":140,"x_max":378,"y_max":172,"width":32...},
     "size":{"radius_pixels":16.5}}
  ]
}
```

### B.3. VisPhyBench Templates and Stochasticity

VisPhyBench templates are defined as executable PHYRE-style task scripts. Unlike static assets, each template is instantiated by sampling seeds (e.g., object placements and sizes), so a single rendered snapshot does not capture the full diversity. We therefore summarize object composition over the full sub split using the detection context $D$ on $I^{\text{start}}$ (see Table 7).

*Table 7.* Object category statistics on VisPhyBench.

| Category | Scenes (%) | Scenes (count) | Objects (count) |
|---|---|---|---|
| circle | 100.0 | 191 | 779 |
| line | 83.2 | 159 | 344 |
| rectangle | 62.8 | 120 | 321 |
| u_shape | 24.6 | 47 | 47 |
| triangle | 6.3 | 12 | 16 |
| composite_shape | 7.3 | 14 | 16 |

**3D templates.** In addition to PHYRE-style 2D scripts, we include a set of programmatic 3D templates implemented in Three.js + Cannon.js. These 3D templates use simple rigid-body primitives (e.g., spheres, boxes, ramps, barriers) under a fixed perspective camera and white background, and are designed to probe depth-aware contacts and occlusions not present in purely 2D scenes. Because $D$ is defined in 2D pixel space from a first-frame detector, the category statistics above are reported for the 2D portion of the split; for the 3D subset we instead rely on the executable template specification and deterministic rendering protocol (Appendix B.4).

### B.4. Deterministic Execution and 2D Constraint

VisPhyWorld executes each generated scene hypothesis under a fixed, deterministic configuration to ensure comparability across models.

**Canonicalization and validation.** Raw model outputs may contain extraneous text or malformed markup. Before execution, we extract the HTML payload (from a fenced ```html block when present, otherwise the outermost <html>...</html> segment), and canonicalize it into a standard executable template that injects the required libraries and a trusted recording helper. We additionally validate basic requirements (e.g., existence of a drawable canvas and finite numeric states). Retry and fallback behaviors are described in Appendix B.5.

**Execution contract.** For each sample, the renderer produces a fixed-length clip $\hat{X}$ at the reference frame rate and duration associated with that sample. All runs use a fixed physics time step and a fixed camera configuration; as a result, variability in $\hat{X}$ is attributable to the generated hypothesis rather than nondeterministic execution.

**2D constraint.** Although the underlying physics engine supports full 3D dynamics, we restrict motion to a 2D plane by (i) initializing all bodies with $z = 0$ and (ii) projecting the state back to the plane at each simulation step (clamping out-of-plane position and angular components to zero). This avoids uncontrolled 3D degrees of freedom while preserving rigid-body contact dynamics.

**3D execution.** For our 3D subset, we disable the 2D clamping rule and execute full 3D rigid-body dynamics with the same deterministic protocol (fixed physics time step, fixed recording duration, and fixed camera parameters). To preserve comparability across models, we keep the camera static and normalize all rendered videos to match the reference FPS, duration, and resolution of the corresponding ground-truth clip.

---

**Deterministic Rendering Protocol (High-Level)**

Given a model-generated scene hypothesis $C$, the renderer produces the output clip $\hat{X}$ under a fixed protocol:

- **Parse & canonicalize:** Extract the HTML payload and wrap it into a standard executable template with fixed library versions and a trusted recorder.

- **Validate:** Check minimal execution requirements (e.g., a drawable canvas and finite numeric states).

- **Execute deterministically:** Run physics with a fixed time step and a fixed orthographic camera, producing frames at the sample's reference FPS.

- **Enforce 2D:** Initialize with $z = 0$ and clamp out-of-plane components each step.

- **Export:** Record a fixed-duration clip and convert it to a standard format for downstream evaluation.

*Figure 17.* High-level deterministic rendering protocol used in VisPhyWorld. Low-level implementation details are included in the released codebase.

### B.5. Robustness: Automatic Retry and Fallback

To handle syntax errors or runtime exceptions in model-generated programs, we implement a lightweight robustness protocol that ensures evaluation is well-defined for all samples.

**Error-conditioned single-step repair.** If the initial program fails to execute (e.g., syntax error, missing canvas, or runtime exception), we capture execution diagnostics (e.g., JavaScript console logs and error traces), summarize them, and provide the summary to the model for a single repair attempt.

**Fallback and well-defined evaluation.** If the repair attempt also fails, we execute a minimal hand-crafted fallback template (Figure 18) that guarantees a valid canvas and finite motion. This prevents missing outputs and ensures the evaluation pipeline does not crash; such samples receive correspondingly poor scores on the metrics.

**Success criteria.** We distinguish two notions of success. **Model-success** counts a sample as successful only if the model-generated hypothesis executes and produces a non-empty clip without invoking the fallback. **System-success** additionally counts fallback clips as successful, and is used only to guarantee that the evaluation pipeline is well-defined. Unless otherwise stated, success rates reported in the main paper use Model-success.

**Fallback Template (Simplified Sketch)**

The fallback template guarantees a valid canvas and finite motion when model generation fails:

```
<!DOCTYPE html>
<html lang="en">
<head>
<meta charset="UTF-8" />
<title>VisPhyWorld Fallback Scene</title>
<script src="three.min.js"></script>
<script src="cannon.min.js"></script>
<script src="recording.js"></script>
</head>
<body style="margin:0;overflow:hidden;">
<canvas id="visphyworld-canvas"></canvas>
<script>
// Setup renderer, camera, scene, lights
// Create a flat ground plane and one spherical body
// Run simulation loop and export a finite-duration clip
</script>
</body>
</html>
```

*Figure 18.* High-level structure of the fallback template used when both model attempts fail.

## C. Evaluation Metrics: Definitions & Protocols

This appendix defines the metric families used in the main paper. All metrics are computed per scene and then averaged over the evaluated split. Unless otherwise noted, frame-wise metrics are computed after temporal alignment (Appendix C.1).

### C.1. Default Evaluation Hyperparameters

We report our default evaluation hyperparameters for reproducibility. Unless otherwise stated, we uniformly sample frames every `sample_every=3` frames for all frame-wise metrics. For temporal alignment, we use a coarse-to-fine strategy with coarse offset search up to `max_offset=30` (in sampled frames), a stack window `window=3`, and offset penalty `offset_penalty=0.05`. The coarse search uses `downsample=64`, `top_k=5`, and `max_samples=16`. When DTW is enabled, we compute frame features using a $48 \times 48$ grayscale thumbnail and a step penalty of `0.005`.

*Table 8.* Default evaluation hyperparameters used throughout the paper.

| Setting | Value |
| --- | --- |
| Frame sampling | `sample_every=3` |
| Coarse offset search | `max_offset=30` (sampled frames) |
| Stack refinement window | `window=3` |
| Offset penalty | `offset_penalty=0.05` |
| Coarse downsample | `downsample=64` |
| Top-$k$ candidates | `top_k=5` |
| Max coarse samples | `max_samples=16` |
| DTW feature size | $48 \times 48$ grayscale |
| DTW step penalty | `0.005` |

### C.2. Reconstruction & Perceptual Quality

- **PSNR and SSIM:** Computed frame-wise between aligned reference and generated videos. SSIM is averaged across the Y channel and RGB channels.

- **LPIPS, FSIM, VSI, DISTS:** Deep and structural perceptual metrics computed on aligned frames. We use the `piq` library implementation.

### C.3. Visual Semantic Consistency

- **CLIP-Img:** Cosine similarity between CLIP (ViT-B/32) embeddings of reference and generated frames, measuring high-level semantic/layout consistency.

- **DINO Similarity:** Cosine similarity of DINO ViT features, which is more sensitive to object structure and less biased by text supervision than CLIP.

### C.4. Text–Video & Analysis Consistency

- **CLIP-Cap:** Similarity between the generated motion-analysis text and the generated video frames.

- **Text Metrics (ROUGE, BERTScore):** We compare the generated analysis against an automatically produced reference description of the original video (generated by a strong LLM). This validates whether the model correctly perceives and verbalizes the events in the input video.

### C.5. Motion & Physical Plausibility

- **RAFT Optical Flow:** We compute End-Point Error (EPE), flow magnitude difference, and angular error between the optical flow fields of the reference and generated videos.

- **Temporal Alignment:** We use a coarse-to-fine alignment strategy (Figure 19) combining offset search and Dynamic Time Warping (DTW) to handle temporal shifts before metric computation.

### C.6. Subjective Quality (Gemini Judge)

We employ **Gemini-2.5-Pro** as a holistic judge. The prompt (Figure 20) asks the model to compare the reference and generated videos and assign a score (1–10) with a justification.

## D. Detailed Experimental Results

We provide the full breakdown of experimental results across all metrics and models.

**Temporal Alignment Procedure**

**(1) Coarse Search:** Downsample frames to grayscale vectors. Compute correlation for offsets $\pm 30$ frames. Keep top-$k$ candidates.
**(2) Stack Refinement:** Build frame stacks ($w = 3$). Minimize cost = MSE + 0.5 MAE + 0.1 Angular.
**(3) DTW:** Run Dynamic Time Warping on low-res features to align variable-speed sequences.

*Figure 19.* Temporal alignment procedure used before computing frame-wise metrics.

**Gemini-based Physics & Video Consistency Prompt**

You are an expert evaluator of **physical simulations** and video quality.
Compare the provided **reference video** (Ground Truth) with the **generated video**. Your goal is to determine if the generated video accurately reconstructs the **physical event** shown in the reference.
Focus on the following dimensions:

- **Physical Plausibility (Crucial):** Do the objects obey rigid-body physics laws (gravity, collisions, friction) in the given setting (2D or 3D)? Are there any "hallucinations" such as objects passing through each other (ghosting), floating unnaturally, or failing to move when hit?

- **Motion Consistency:** Does the trajectory, speed, and timing of the movement align with the reference?

- **Scene Semantics:** Are the correct objects (color, shape, count) present in the correct layout?

- **Visual Fidelity:** Overall clarity, ignoring minor rendering style differences if the physics is correct.

Return a JSON object with the keys:

- `score`: integer between 1 and 10.

  - **10**: Perfect physical and visual match.
  - **1**: Physical laws are violated (e.g., phantom collision, static scene when motion is expected), even if the image looks realistic.

- `justification`: Brief explanation, specifically pointing out any physical violations if present.

*Figure 20.* Prompt template used for Gemini-based evaluation. Note that the prompt is explicitly designed to penalize **physical violations** (e.g., incorrect collision logic), ensuring the score reflects physical understanding rather than just perceptual similarity.

### D.1. VisPhyBench Difficulty Stratification

Table 3 in the main paper reports the difficulty distribution; here we provide additional details on the stratification and split construction. All annotators are graduate students with STEM backgrounds.

### D.2. Per-Scene Distributions and Significance (Sub Split)

Mean scores can obscure whether improvements are driven by a small subset of scenes. To address this, we report (i) per-scene metric distributions via boxplots (Figure 21) and (ii) paired bootstrap confidence intervals over per-scene differences (Table 9). We use paired resampling because all methods are evaluated on the same set of scenes ($N = 209$), and define "mean improvement" so that positive values indicate better performance by VisPhyWorld (GPT-5, threejs), taking metric direction into account ($\uparrow / \downarrow$).

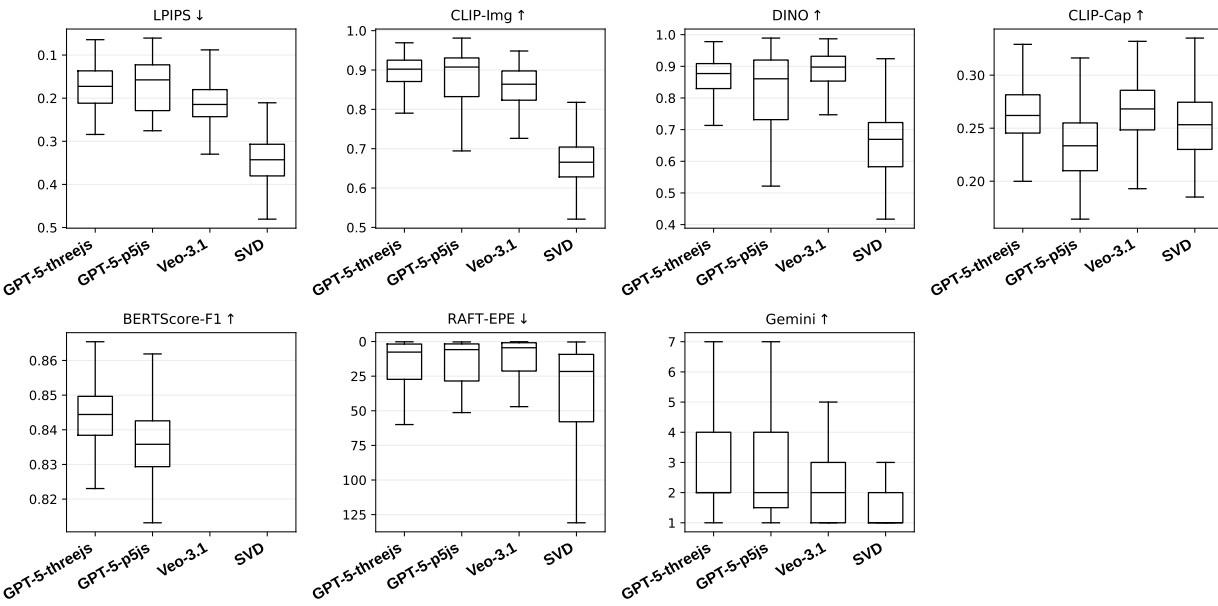

*Figure 21.* Per-scene boxplot distributions on VisPhyBench for representative metric families (higher is better unless marked ↓).

### D.3. Reconstruction & Perceptual Metrics

Table 10 details the pixel-level and perceptual metrics. Gemini-3-Pro consistently achieves the best perceptual scores (LPIPS, FSIM), while Three.js backends generally outperform P5.js.

### D.4. Visual Semantic Consistency

Table 11 compares semantic understanding. GPT-5 and Gemini-3-Pro show strong alignment with the ground truth in terms of CLIP and DINO scores.

### D.5. Text & Physical Consistency

Table 12 and Table 13 (below) provide the remaining metrics on text analysis quality and physical motion fidelity.

| Metric | Comparison | Mean improvement | 95% bootstrap CI |
|---|---|---|---|
| LPIPS↓ | VisPhyWorld (GPT-5, p5js) | 0.1143 | [0.0743, 0.1567] |
| LPIPS↓ | Veo-3.1 | 0.0365 | [0.0310, 0.0420] |
| LPIPS↓ | SVD (img2vid) | 0.1674 | [0.1581, 0.1764] |
| CLIP-Img↑ | VisPhyWorld (GPT-5, p5js) | 0.0754 | [0.0477, 0.1044] |
| CLIP-Img↑ | Veo-3.1 | 0.0365 | [0.0285, 0.0446] |
| CLIP-Img↑ | SVD (img2vid) | 0.2258 | [0.2159, 0.2355] |
| DINO↑ | VisPhyWorld (GPT-5, p5js) | 0.0957 | [0.0643, 0.1290] |
| DINO↑ | Veo-3.1 | -0.0276 | [-0.0340, -0.0217] |
| DINO↑ | SVD (img2vid) | 0.2036 | [0.1917, 0.2155] |
| CLIP-Cap↑ | VisPhyWorld (GPT-5, p5js) | 0.0299 | [0.0243, 0.0356] |
| CLIP-Cap↑ | Veo-3.1 | -0.0050 | [-0.0098, -0.0002] |
| CLIP-Cap↑ | SVD (img2vid) | 0.0101 | [0.0050, 0.0151] |
| BERTScore-F1↑ | VisPhyWorld (GPT-5, p5js) | 0.0077 | [0.0059, 0.0094] |
| BERTScore-F1↑ | Veo-3.1 | N/A | [N/A, N/A] |
| BERTScore-F1↑ | SVD (img2vid) | N/A | [N/A, N/A] |
| RAFT-EPE↓ | VisPhyWorld (GPT-5, p5js) | 0.6294 | [-0.7743, 2.0695] |
| RAFT-EPE↓ | Veo-3.1 | -0.7078 | [-1.8371, 0.4628] |
| RAFT-EPE↓ | SVD (img2vid) | 11.9706 | [8.8544, 15.1865] |
| Gemini↑ | VisPhyWorld (GPT-5, p5js) | -0.0108 | [-0.5081, 0.5027] |
| Gemini↑ | Veo-3.1 | 0.9153 | [0.4233, 1.3968] |
| Gemini↑ | SVD (img2vid) | 2.0635 | [1.7090, 2.4444] |

*Table 9.* Paired bootstrap confidence intervals (VisPhyBench `sub`, $N = 209$). "Mean improvement" is defined so that positive values indicate VisPhyWorld (GPT-5, threejs) performs better (for ↓ metrics we compute baseline−ours; for ↑ metrics ours−baseline).

*Table 10.* Detailed breakdown of Reconstruction and Perceptual Metrics.

| Model | PSNR↑ | SSIM↑ | LPIPS↓ | FSIM↑ | VSI↑ | DISTS↓ |
|---|---|---|---|---|---|---|
| VisPhyWorld (GPT-5, threejs) | 20.54 | 0.9370 | 0.1736 | 0.9014 | 0.8432 | 0.1883 |
| VisPhyWorld (GPT-5, p5js) | 16.36 | 0.7440 | 0.2926 | 0.9105 | 0.8193 | 0.2724 |
| VisPhyWorld (GPT-4.1, threejs) | 19.74 | 0.9337 | 0.1818 | 0.9064 | 0.8309 | 0.2040 |
| VisPhyWorld (GPT-4.1, p5js) | 14.83 | 0.6830 | 0.3520 | 0.8977 | 0.8112 | 0.3348 |
| VisPhyWorld (Gemini-3-Pro, threejs) | **21.26** | **0.9445** | **0.1399** | **0.9225** | **0.8539** | 0.1859 |
| VisPhyWorld (Gemini-3-Pro, p5js) | 15.57 | 0.6943 | 0.3302 | 0.9055 | 0.8220 | 0.3384 |
| VisPhyWorld (Claude Sonnet 4.5, threejs) | 20.75 | 0.9406 | 0.1602 | 0.9118 | 0.8374 | 0.2001 |
| VisPhyWorld (Claude Sonnet 4.5, p5js) | 15.36 | 0.7160 | 0.3250 | 0.9030 | 0.8162 | 0.3109 |
| VisPhyWorld (Qwen3-VL-Plus, threejs) | 18.66 | 0.9306 | 0.2207 | 0.8972 | 0.8099 | 0.2373 |
| VisPhyWorld (Qwen3-VL-Plus, p5js) | 9.14 | 0.4296 | 0.5478 | 0.8797 | 0.7886 | 0.4396 |
| SVD (img2vid) | 14.44 | 0.8802 | 0.3408 | 0.8239 | 0.7585 | 0.3459 |
| Veo-3.1 | 20.04 | 0.9354 | 0.2102 | 0.8561 | 0.8586 | **0.1755** |

*Table 11.* Visual Semantic Consistency Metrics.

| Model | CLIP-Img↑ | DINO↑ |
|---|---|---|
| VisPhyWorld (GPT-5, threejs) | 0.8930 | 0.8556 |
| VisPhyWorld (GPT-5, p5js) | 0.8134 | 0.7580 |
| VisPhyWorld (GPT-4.1, threejs) | 0.8933 | 0.8304 |
| VisPhyWorld (GPT-4.1, p5js) | 0.7545 | 0.6786 |
| VisPhyWorld (Gemini-3-Pro, threejs) | **0.8973** | 0.8405 |
| VisPhyWorld (Gemini-3-Pro, p5js) | 0.7460 | 0.6721 |
| VisPhyWorld (Claude Sonnet 4.5, threejs) | 0.8957 | 0.8305 |
| VisPhyWorld (Claude Sonnet 4.5, p5js) | 0.7612 | 0.7098 |
| VisPhyWorld (Qwen3-VL-Plus, threejs) | 0.8717 | 0.7837 |
| VisPhyWorld (Qwen3-VL-Plus, p5js) | 0.6446 | 0.5478 |
| SVD (img2vid) | 0.6677 | 0.6528 |
| Veo-3.1 | 0.8564 | **0.8839** |

*Table 12.* Text–Video and Analysis-Text Consistency Metrics.

| Model | CLIP-Cap↑ | ROUGE-L F1↑ | BERTScore-F1↑ |
|---|---|---|---|
| VisPhyWorld (GPT-5, threejs) | 0.2632 | 0.2186 | 0.8436 |
| VisPhyWorld (GPT-5, p5js) | 0.2331 | 0.2057 | 0.8360 |
| VisPhyWorld (GPT-4.1, threejs) | 0.2610 | **0.2383** | **0.8522** |
| VisPhyWorld (GPT-4.1, p5js) | 0.2192 | 0.1689 | 0.8253 |
| VisPhyWorld (Gemini-3-Pro, threejs) | 0.2567 | 0.2141 | 0.8460 |
| VisPhyWorld (Gemini-3-Pro, p5js) | 0.2184 | 0.1886 | 0.8396 |
| VisPhyWorld (Claude Sonnet 4.5, threejs) | 0.2588 | 0.2168 | 0.8468 |
| VisPhyWorld (Claude Sonnet 4.5, p5js) | 0.2177 | 0.1599 | 0.8224 |
| VisPhyWorld (Qwen3-VL-Plus, threejs) | **0.2650** | 0.2022 | 0.8466 |
| VisPhyWorld (Qwen3-VL-Plus, p5js) | 0.2032 | 0.1733 | 0.8358 |
| SVD (img2vid) | 0.2533 | – | – |
| Veo-3.1 | **0.2681** | – | – |

*Table 13.* Motion and Physical Plausibility Metrics (Selected columns).

| Model | RAFT-EPE↓ | RAFT-Angle↓ | Align-Err↓ |
|---|---|---|---|
| VisPhyWorld (GPT-5, threejs) | 33.6473 | 68.5500 | 0.0210 |
| VisPhyWorld (GPT-5, p5js) | 34.3433 | 75.8555 | 0.0279 |
| VisPhyWorld (GPT-4.1, threejs) | 33.7110 | 67.7974 | 0.0249 |
| VisPhyWorld (GPT-4.1, p5js) | 37.6993 | 82.9492 | 0.0397 |
| VisPhyWorld (Gemini-3-Pro, threejs) | 36.2030 | **62.4494** | 0.0192 |
| VisPhyWorld (Gemini-3-Pro, p5js) | 33.1013 | 81.5723 | **0.0184** |
| VisPhyWorld (Claude Sonnet 4.5, threejs) | 36.1985 | 71.7979 | 0.0210 |
| VisPhyWorld (Claude Sonnet 4.5, p5js) | 34.1425 | 78.2841 | 0.0277 |
| VisPhyWorld (Qwen3-VL-Plus, threejs) | 35.0493 | 75.6650 | 0.0350 |
| VisPhyWorld (Qwen3-VL-Plus, p5js) | **20.8187** | 80.7413 | 0.8567 |
| SVD (img2vid) | 45.4606 | 84.7314 | 0.0746 |
| Veo-3.1 | 32.7145 | 77.0550 | 0.0193 |

