# OpenReview forum: "VisPhyWorld: Probing Physical Reasoning via Code-Driven Video Reconstruction"
_ICML.cc/2026/Conference — Submitted to ICML 2026_

### Official Review · Reviewer_mDvZ · 2026-02-15

**Soundness:** 3
**Presentation:** 2
**Significance:** 1
**Originality:** 2
**Overall Recommendation:** 3
**Confidence:** 4

**Summary:**

The paper proposes a framework for evaluating the physical reasoning capabilities of MLLMs. They introduced an Executable hypothesis paradigm: instead of predicting future pixels directly, the model must generate executable simulation code to reconstruct a scene given two key frames. The generated code will be rendered into a video and evaluated against the ground truth. For evaluation, they proposed VisPhyBench, which is a dataset of 209 scenes.

**Compliance With Llm Reviewing Policy:**

Affirmed.

**Final Justification:**

na

**Key Questions For Authors:**

1. Have you tried to use a different specialized model as the coding agent to mitigate the bias from coding ability?
2. In the "No Retry" vs. "1 Retry" ablation, you show a significant jump in success rates. Does this suggest that the primary bottleneck for some models is syntax/API compliance rather than actual physical reasoning failures?
3. Did you perform any human evaluation on a subset of the data to validate the reliability of the Gemini-2.5-Pro judge?
4. Evaluating physical understanding is more needed for world models. Have you evaluated more specific world models like Cosmos [1] on your benchmark?


[1] Niket Agarwal, Arslan Ali, Maciej Bala, Yogesh Balaji, Erik Barker, Tiffany Cai, Prithvijit Chattopadhyay, Yongxin Chen, Yin Cui, Yifan Ding, et al. Cosmos world foundation model platform for physical AI. arXiv preprint arXiv:2501.03575, 2025

**Limitations:**

yes

**Strengths And Weaknesses:**

### **Strengths**:
1. By forcing the model to explicitly parameterize the scene (mass, friction, velocity) in code, the framework decouples reasoning from rendering, which is useful in some hybrid scenarios
2. Unlike black-box video generation, where errors appear as visual artifacts, VisPhyWorld allows inspection of the generated code to pinpoint specific reasoning failures

### **Weaknesses**:
1. Limited scope & synthetic domain: The benchmark is restricted to simple, synthetic rigid-body primitives, which are far different from physical understanding in real-world scenarios. This limits the claims regarding "general" physical reasoning.
2. Conflation of Coding vs. Reasoning:  The model's coding ability is thoroughly affecting the evaluation of physical understanding. A model with strong physical intuition but poor syntax knowledge (e.g., forgetting a Three.js boilerplate) is penalized heavily. A two-stage pipeline (reasoning agent $\to$ coding agent) might have mitigated this bottleneck. This will also limit the scalability of the benchmark because if we add more dynamics and complexity, then the outcome code would be much more complex, which will enlarge the gap between coding and physical understanding.
3. Reliability of the VLM Judge: The reliance on a Gemini-2.5-Pro judge for the Holistic quality metric is a significant methodological risk. While the authors argue this captures semantic failures that RAFT misses, there is no human-validation study provided to calibrate the judge. Furthermore, the fact that the Gemini judge assigned the highest score to the Gemini 3 Pro model raises concerns about self-preference bias.
4. Scale: the benchmark size (only 209 scenes from 108 templates) seems relatively small for a benchmark intended to evaluate "general" physical reasoning! It is unclear how this approach translates to complex, real-world scenes.
5. Error analysis: a comprehensive error analysis is missing to show which part models mostly struggle with in physical reasoning
6. minor typos and edits: text is collapsing into diagram in figure 4 making it hard to read. Equation 1 should end with dot not comma. Figure 2 and 3 has low resolution (use vector files like svg). The first paragraph of related work is too long making it hard to follow.

---

> ### Author Rebuttal · Authors · 2026-03-31
>
> **W1**: Limited scope…
>
> We agree that VisPhyBench currently focuses mainly on controlled synthetic rigid-body scenes. We view this benchmark as an early exploration of this problem setting, which is also consistent with prior works such as CoPhy [1] and PhyWorld [2], which use relatively simple and controlled scenes.Current MLLMs often cannot reliably identify object types, positions, and layouts, making it difficult to separate perceptual failures from physical reasoning failures.
>
> **W2**: Conflation of Coding…
>
> Our goal is not to separate physical reasoning from coding, but to test whether models can express physical understanding through executable simulation code. Since many MLLMs cannot directly generate videos, code provides a practical evaluation interface. To reduce toolchain bias, we evaluate multiple backends and also include direct video-generation baselines such as Sora-2 and Veo-3.1.
>
> We appreciate the reviewer’s suggestion. However, we argue that a two-stage pipeline would likely make the problem harder rather than easier. In the first stage, the reasoning agent must convert a visual scene into a natural-language description, which inevitably loses spatial precision: object positions, sizes, and initial velocities are difficult to express accurately in free text. The coding agent in the second stage then receives only this imprecise description and has no access to the original visual input, making it harder to produce a faithful simulation. Our one-stage approach allows the model to directly translate visual understanding into code while retaining full scene detail throughout.
>
> **W3 & Q3**: Reliability…
>
>
> Thank you for this question. We replace the judge model, and further conduct human evaluation. Although exact rankings vary, the overall conclusions remain stable: inter-judge correlations are significant, weak baselines such as SVD-img2vid consistently rank at the bottom. This supports the robustness of our conclusions to both auxiliary context removal and judge replacement.
>
>
> 1. Different judge models:
>
> |Model|Gemini score|GPT-5.4 score|Qwen3-VL score|
> |---|---:|---:|---:|
> |Veo-3.1|2.56|5.09|7.82|
> |Gemini-3-Pro|3.81|3.99|6.74|
> |GPT-5|3.53|3.33|7.12|
> |Claude-Sonnet-4.5|2.50|3.60|6.52|
> |Qwen3-VL-Plus|2.13|2.25|4.94|
>
>
> 2. Sample-level judge agreement correlation table(all p<0.001):
>
> |Judge Pair|Pearson r|Spearman ρ|
> |---|---:|---:|
> |Gemini vs GPT-5.4|0.41|0.37|
> |Gemini vs Qwen3-VL|0.40|0.43|
> |GPT-5.4 vs Qwen3-VL|0.65|0.70|
>
> We also conduct a human evaluation with 6 STEM graduate students. Results show that the automated metrics are broadly aligned with human perception.
>
> |Model|Human Evaluation Score|
> |---|---:|
> |Ours(Gemini-3-Pro)|4.67|
> |Veo-3.1|4.50|
> |Ours(GPT-5)|4.33|
> |Ours(GPT-4.1)|4.17|
> |Ours(Claude Sonnet 4.5)|3.83|
> |Sora 2|2.83|
> |Ours(Qwen3-VL-Plus)|2.50|
> |SVD(img2vid)|1.83|
> |Cosmos|1.16|
>
>
> **W4**: Scale.
>
> We currently focus on controlled synthetic rigid-body scenes as an early but necessary step. Our 2D subset is not directly borrowed from PhyWorld: although both use the PHYRE engine, ours contains 108 templates ( PhyWorld has 70 templates) with more shapes, layouts, and variations. This scale is also aligned with prior work such as PhyWorld[2], and Physics-IQ[7], and helps reduce the confound between perceptual failures and physical reasoning failures in more complex scenes.
>
> **W5**: Error analysis.
>
> Our paper already includes qualitative error analysis in Section 4.2 Case Study and Figures 5, 6, and 7, with additional examples in the appendix. These examples show a consistent pattern: models are often able to capture coarse scene semantics, but they struggle more with the physical quantities and dynamics.
>
> **W6**: Minor typos…
>
> Thank you for comments. We will fix all of these issues in the revision.
>
> **Q1**: Have you tried…
>
> To our knowledge, there is currently no established coding agent for this video-to-code-video simulation reconstruction. We already use very strong frontier MLLM, so the bottleneck is not simply the absence of a stronger coding model.
>
> **Q2**: In the "No Retry"....
> We want to clarify that failed samples that do not produce a video are not included in our downstream quality metrics, so retry affects executability, not the evaluation of physical correctness itself.
>
> **Q4**：More baselines.
>
> we further included Cosmos and Sora-2:
>
> |Model|LPIPS↓|CLIP-Img↑|DINO↑|CLIP-Cap↑|BERTScore-F1↑|RAFT-EPE↓|Gemini↑|Human Eval↑|
> |---|---:|---:|---:|---:|---:|---:|---:|---:|
> |Sora-2|0.20|0.87|0.87|0.26|--|34.91|2.35|2.83|
> |Cosmos-Predict2.5-2B†|0.52|0.63|0.54|--|--|30.91|1.16|1.16|
>
> Sora-2 still lag in physical correctness, while Cosmos performs much worse overall; in human evaluation, we observe that Cosmos often fails to maintain recognizable objects. These suggest even with frontier models, lacks the transparency and controllability of code-based simulation for evaluating physical reasoning.
>
> **References:** See anonymous.4open.science/r/VisPhyWorld-rebuttal-B6BF/README.md

---

> > ### Author Rebuttal · Reviewer_mDvZ · 2026-04-05
> >
> > I thank the author for their answers. I have increased my score since some other concern of mine got resolved.
> >
> > Even if you are trying to evaluate whether the model can express physical understanding through executable simulation code, you should still try to separate code generation ability and physical understanding of the model. Because, as you mentioned in the introduction, the main question is whether models understand physics and whether they can generate physically aware output or not, and you are using code generation only as a tool here!

---

> > > ### Author Response · Authors · 2026-04-07
> > >
> > > Thank you for the helpful follow-up. We agree that, since code generation is only the interface and not the final goal, it is important to check whether our results are mainly driven by backend-specific coding ability.
> > >
> > > To test this directly, we added a new multi-engine study using the same scenes and the same model, GPT-5, across five different backends: Three.js/JavaScript, P5.js, SVG, Manim/Python, and Blender/bpy. These backends require different languages and APIs. If the results were mainly driven by syntax knowledge or backend-specific coding knowledge, then scene-level performance should change a lot across engines. But we find the opposite. On the most important physics-related metric, RAFT-EPE, the scene ranking is very consistent across engines, with an average Spearman correlation of 0.802 across all engine pairs, and all pairwise correlations are significant, with p < 0.05. In comparison, appearance and semantic metrics are only moderately correlated across engines, with average correlations of 0.427 for LPIPS and 0.466 for DINO.
> > >
> > > We believe this is the key finding. Changing the backend does affect the overall difficulty, but it does not strongly change which scenes the model does well or poorly on for motion dynamics.
> > >
> > >
> > > Table 1: Absolute cross-engine results. While absolute scores vary by backend, the scene-level motion ranking remains highly consistent across engines.
> > >
> > > |Engine|LPIPS↓|CLIP-Img↑|DINO↑|CLIP-Cap↑|BERTScore↑|RAFT-EPE↓|
> > > |---|---:|---:|---:|---:|---:|---:|
> > > |Three.js|0.15|0.89|0.86|0.26|0.84|27.09|
> > > |SVG|0.17|0.92|0.87|0.26|0.85|29.55|
> > > |Blender|0.24|0.75|0.71|0.30|0.84|27.73|
> > > |P5.js|0.29|0.80|0.74|0.23|0.81|33.92|
> > > |Manim|0.29|0.85|0.79|0.25|0.84|35.79|
> > >
> > > Table 2: Cross-engine Spearman correlation matrix on RAFT-EPE. (Average ρ=0.80, * indicates p < 0.05.)
> > >
> > > |RAFT-EPE↓|ThreeJS|P5.js|SVG|Manim|Blender|
> > > |---|---:|---:|---:|---:|---:|
> > > |ThreeJS|1.00|||||
> > > |P5.js|0.64*|1.00||||
> > > |SVG|0.90*|0.79*|1.00|||
> > > |Manim|0.71*|0.94*|0.83*|1.00||
> > > |Blender|0.64*|0.91*|0.79*|0.86*|1.00|
> > >
> > > Table 3: Cross-engine Spearman correlation matrix on LPIPS. (Average ρ=0.43)
> > >
> > > |LPIPS↓|ThreeJS|P5.js|SVG|Manim|Blender|
> > > |---|---:|---:|---:|---:|---:|
> > > |ThreeJS|1.00|||||
> > > |P5.js|0.24|1.00||||
> > > |SVG|0.55*|0.65*|1.00|||
> > > |Manim|0.38|0.57*|0.41|1.00||
> > > |Blender|0.14|0.48*|0.45|0.40|1.00|
> > >
> > > Table 4: Cross-engine Spearman correlation matrix on DINO. (Average ρ=0.47)
> > >
> > > |DINO↑|ThreeJS|P5.js|SVG|Manim|Blender|
> > > |---|---:|---:|---:|---:|---:|
> > > |ThreeJS|1.00|||||
> > > |P5.js|0.17|1.00||||
> > > |SVG|0.65*|0.24|1.00|||
> > > |Manim|0.53*|0.42|0.29|1.00||
> > > |Blender|0.55*|0.68*|0.49*|0.64*|1.00|
> > >
> > >
> > >
> > > As noted in our previous response, we use code as the intermediate representation because most currently available APIs do not directly provide the kind of image or video conditioned video generation interface required by our benchmark, including the API-accessible versions of Claude Sonnet 4.5, GPT-5, Gemini 3 Pro, and Qwen3-VL-Plus. To reduce dependence on any single toolchain, we experiment with multiple rendering backends, including Three.js, P5.js, SVG, Manim, and Blender, and also include direct video generation baselines such as Sora-2 and Veo-3.1.  Since Sora-2 and Veo-3.1 bypass code as the intermediate representation, they serve as an additional check that our conclusions are not solely driven by a code-based pipeline.
> > >
> > >
> > > More broadly, our design follows a common pattern in multimodal generation: many systems separate high-level understanding from final visual synthesis, often by using a VLM/LLM-like module to produce semantic conditions and a diffusion decoder to generate pixels. In our setting, executable simulation code plays a similar role as an intermediate representation. Unlike diffusion decoders, however, code is directly inspectable, editable, and physically falsifiable. We do not claim to completely isolate reasoning from implementation; rather, our new multi-engine results show that backend-specific coding knowledge is not the dominant driver of benchmark outcomes. [1-3]
> > >
> > > [1]: Mi et al. ThinkDiff: I Think, Therefore I Diffuse: Enabling Multimodal In-Context Reasoning in Diffusion Models. ICML 2025
> > >
> > > [2]: Yang et al. FOCUS: Unified Vision-Language Modeling for Interactive Editing Driven by Referential Segmentation. NeurIPS 2025
> > >
> > > [3]: Zhao et al. EasyGen: Easing Multimodal Generation with BiDiffuser and LLMs. ACL 2024.

---

### Official Review · Reviewer_2ZXA · 2026-03-10

**Soundness:** 2
**Presentation:** 3
**Significance:** 2
**Originality:** 3
**Overall Recommendation:** 4
**Confidence:** 4

**Summary:**

VisPhyWorld proposes an execution-based evaluation framework for probing physical reasoning in MLLMs by requiring models to generate executable simulator code from two video keyframes, then rendering that code to produce a reconstructed video. The authors introduce VisPhyBench (209 scenes, 108 templates, 2D and 3D) and evaluate five frontier MLLMs under a multi-metric protocol. The central finding is that current models are competent at semantic scene parsing but struggle to infer physically correct dynamics and parameterize scenes consistently. The paper positions code generation as a shift from recognition-based to hypothesis-based evaluation, distinguishing itself from prior work (notably PhyWorld) by exposing symbolic intermediate artifacts rather than only rendered video output.

**Compliance With Llm Reviewing Policy:**

Affirmed.

**Final Justification:**

The paper presents a novel and useful evaluation framework with clear presentation and practical significance, though some limitations remain in soundness due to indirect evaluation and benchmark scale. The rebuttal addressed my main concerns with additional experiments and clarifications, which increased my confidence, so I maintain my recommendation.

**Key Questions For Authors:**

**Q1:** Since VisPhyBench is generated from templated simulators with known ground-truth physical parameters, could the authors extract inferred values such as initial velocity, mass, restitution, or gravity from the generated code and report quantitative parameter-level error against ground truth? This would directly operationalize the central claim that models struggle with physical parameterization, and such analysis could substantially strengthen the empirical conclusions.

**Q2:** The default input provides both $I_{start}$ and $I_{later}$, which may allow models to perform endpoint-matching rather than genuine forward physical prediction; could the authors report results for a forecast-only setting using only $I_{start}$ and quantify the effect on motion plausibility metrics, as a meaningful performance gap between the two conditions would clarify whether the framework is genuinely testing physical forecasting ability.

**Q3:** The detection context $D$ is obtained from GPT-5.2 and provided to all evaluated models; could the authors report an ablation comparing performance with and without this context (or with a simpler detector) for at least two models? This would clarify how much the results depend on this additional preprocessing stage and help isolate the models’ intrinsic reasoning ability.

**Q4:**  Given that Gemini-2.5-Pro serves as the holistic judge while Gemini-3-Pro is one of the evaluated generators, could the authors report correlation between the Gemini judge scores and either human rater assessments or an alternative VLM judge such as GPT-4V on a subset of scenes, as this calibration would strengthen confidence in the holistic score as an unbiased evaluation signal.

**Limitations:**

Yes

**Strengths And Weaknesses:**

**Strengths**

- The paper introduces a novel evaluation framing by requiring models to produce executable simulator code rather than answers or generated videos. This “executable hypothesis” setup creates an explicit interface between perception and simulation and exposes intermediate artifacts (analysis text, scene specification, and code) that make the model’s inferred world state more transparent than pixel-only outputs.

- The paper is generally clearly written and well structured. The pipeline from visual input → code generation → re-simulation is easy to follow, and the case studies and prompt templates help concretely illustrate the framework across models and backends.

- The work addresses an important and timely question: how to evaluate physical reasoning capabilities of multimodal models beyond recognition-style benchmarks. The backend comparison (e.g., Three.js vs P5.js vs SVG/Manim) provides practical insights about how physics-enabled engines influence reconstruction quality.

- The self-repair ablation is a useful diagnostic result showing that many failures arise from superficial code-generation issues rather than deeper reasoning failures, suggesting the execution pipeline is reasonably robust on this benchmark.

**Weaknesses**

- The central claim that executable code exposes inspectable physical hypotheses is not operationalized in the evaluation. All reported metrics compare rendered videos against ground truth rather than evaluating the physical parameters embedded in the generated code (e.g., velocity, friction, restitution) against simulator ground truth.

- The default input includes both an initial frame and a later frame, which may allow models to match the final state rather than perform genuine forward prediction. Without a forecast-only setting using only the initial frame, it is difficult to determine whether the framework measures physical forecasting or reconstruction consistency.

- The evaluation relies on indirect or potentially biased signals for motion and plausibility. Motion quality is evaluated using RAFT-based optical flow metrics and a proprietary LLM judge; additionally, Gemini-2.5-Pro serves as the holistic judge while Gemini-3-Pro is one of the evaluated generators, and no calibration against human raters or alternative judges is provided.

- Several evaluation details could be clarified. The abstract’s 97.7% success claim does not specify whether it reflects model-success or system-success as defined in the appendix, and the Gemini judge scores (max ~3.80 on a nominally 1–10 scale) are not explained in the main text.

- From a significance and originality perspective, the benchmark is relatively small and synthetic (209 scenes, with a small test split and very few hard examples), which limits statistical reliability and generalization claims. The entire 2D subset is also borrowed directly from PhyWorld, which constrains the dataset novelty claim.

---

> ### Author Rebuttal · Authors · 2026-03-31
>
> **W1 & Q1**: The central claim…
>
> Thank you for the suggestion. We additionally conduct a direct analysis to assess the physical parameter correctness via MAE between GT and predicted parameters.
>
> |Parameter|GT mean|GPT-5 MAE|Gemini-3-Pro MAE|Claude-S4-5 MAE|
> |---|---:|---:|---:|---:|
> |Gravity|9.82 m/s²|0|0|0|
> |Restitution|0.15|0.17±0.14|0.18±0.13|0.16±0.05|
> |Friction|0.40|0.10±0.07|0.11±0.07|0.10±0.03|
> |Dynamic mass|1.25 kg|0.15±0.17|0.59±0.88|0.26±0.42|
> |Init speed|2.59 m/s|2.07±2.69|1.69±2.42|2.39±3.23|
>
> These results are consistent with our other metrics: although Gemini-3-Pro shows strong visual reconstruction, it still makes large errors on dynamic mass and initial speed. This reinforces our core claim that current models can generate visually plausible videos yet still fail to recover the quantitative physical parameters governing the dynamics.
>
> **W2 & Q2**: Ablation on forecast-only.
>
> Thank you for this insightful question. We conducted a forecast-only ablation, providing only frame₁ (no frame₁₀).
>
> |Model|Mode|LPIPS↓|CLIP-Img↑|DINO↑|CLIP-Cap↑|BERTScore↑|RAFT-EPE↓|Gemini↑|
> |---|---|---:|---:|---:|---:|---:|---:|---:|
> |GPT-5|frame₁+frame₁₀|0.30|0.87|0.81|0.27|0.85|28.23|2.70|
> |GPT-5|forecast-only|0.31|0.87|0.81|0.28|0.85|26.94|1.60|
> |Gemini-3-Pro|frame₁+frame₁₀|0.22|0.90|0.84|0.25|0.85|30.28|2.90|
> |Gemini-3-Pro|forecast-only|0.21|0.89|0.85|0.25|0.85|27.40|2.45|
> |Claude-S4.5|frame₁+frame₁₀|0.27|0.89|0.81|0.26|0.86|28.46|1.95|
> |Claude-S4.5|forecast-only|0.27|0.88|0.81|0.26|0.87|28.31|1.65|
>
> The forecast-only ablation suggests that models are not simply matching the later frame. When frame₁₀ is removed, visual metrics change little and RAFT-EPE even slightly improves, but the Gemini physical-plausibility score drops more clearly. This indicates that frame₁₀ mainly provides useful physical context, rather than serving as a shortcut endpoint to copy. Without it, models can still generate visually reasonable motion, but are less likely to recover the correct physical outcome.
>
> **W3 & Q3 & Q4**: The evaluation relies…
>
> Thank you for this question. Without detection context, all tested models fall to similarly low holistic scores, suggesting that the context mainly helps object discovery and initialization rather than physical reasoning.
>
> We also replace the judge model, and further conduct human evaluation. Although exact rankings vary, the overall conclusions remain stable: inter-judge correlations are significant, weak baselines such as SVD-img2vid consistently rank at the bottom. This supports the robustness of our conclusions to both auxiliary context removal and judge replacement.
>
>
>
> 1. Different judge models:
>
> |Model|Gemini score|GPT-5.4 score|Qwen3-VL score|
> |---|---:|---:|---:|
> |Veo-3.1|2.56|5.09|7.82|
> |Gemini-3-Pro|3.81|3.99|6.74|
> |GPT-5|3.53|3.33|7.12|
> |GPT-4.1|3.04|3.33|6.84|
> |Claude-Sonnet-4.5|2.50|3.60|6.52|
> |Qwen3-VL-Plus|2.13|2.25|4.94|
> |SVD-img2vid|1.44|1.42|2.92|
>
>
> 2. Model-level scores without detection context
>
> |Model|Mode|LPIPS↓|CLIP-Img↑|DINO↑|CLIP-Cap↑|BERTScore↑|RAFT-EPE↓|Gemini↑|
> |---|---|---:|---:|---:|---:|---:|---:|---:|
> |GPT-5|with detection|0.17|0.89|0.86|0.26|0.84|33.65|3.50|
> |GPT-5|no detection|0.29|0.88|0.81|0.28|0.85|30.55|1.98|
> |Gemini-3-Pro|with detection|0.14|0.90|0.84|0.26|0.85|36.20|3.80|
> |Gemini-3-Pro|no detection|0.21|0.90|0.84|0.25|0.85|31.77|1.84|
> |Claude Sonnet 4.5|with detection|0.16|0.90|0.83|0.26|0.85|36.20|2.39|
> |Claude Sonnet 4.5|no detection|0.27|0.89|0.82|0.25|0.86|31.80|1.65|
>
>
> We also conduct a human evaluation with 6 STEM graduates, who rated generated videos against ground truth in randomized, model-blind order on a 5-point scale. The human ranking shows the same overall trend, suggesting broad alignment between the automated metrics and human perception.
>
> |Model|Human Evaluation Score|
> |---|---:|
> |Ours(Gemini-3-Pro,threejs)|4.67|
> |Veo-3.1|4.50|
> |Ours(GPT-5,threejs)|4.33|
> |Ours(GPT-4.1,threejs)|4.17|
> |Ours(Claude Sonnet 4.5,threejs)|3.83|
> |Sora 2|2.83|
> |Ours(Qwen3-VL-Plus,threejs)|2.50|
> |SVD(img2vid)|1.83|
> |Cosmos|1.16|
>
>
> **W4**: Several evaluation…
>
> Thank you for pointing this out. We will clarify it in revision. The 97.7% in the abstract refers to the full pipeline success rate, not per-model success rate, and the Gemini holistic score is on a 1--10 scale (1 = worst). The top score of about 3.8 reflects the benchmark difficulty, not any hidden rescaling.
>
> **W5**: From a significance…
>
> We currently focus on controlled synthetic rigid-body scenes as an early but necessary step. Our 2D subset is not directly borrowed from PhyWorld: although both use the PHYRE engine, ours contains 108 templates ( PhyWorld 70 templates) with more shapes, layouts, and variations. This scale is also aligned with prior work such as PhyWorld[2], and Physics-IQ[7], and helps reduce the confound between perceptual failures and physical reasoning failures in more complex scenes.
>
> **References:** See anonymous.4open.science/r/VisPhyWorld-rebuttal-B6BF/README.md

---

> > ### Author Rebuttal · Reviewer_2ZXA · 2026-04-04
> >
> > I appreciate the authors' clarifications. My assessment of the paper remains the same, and I will maintain my positive score.

---

> > > ### Author Response · Authors · 2026-04-07
> > >
> > > We sincerely thank the reviewer for the constructive feedback and for recognizing the value of our work.

---

### Official Review · Reviewer_mREd · 2026-03-12

**Soundness:** 3
**Presentation:** 3
**Significance:** 3
**Originality:** 4
**Overall Recommendation:** 4
**Confidence:** 4

**Summary:**

This paper proposes `VisPhyWorld`, an evaluation framework aimed at testing whether MLLMs can perform physical reasoning by requiring them to generate executable simulation code from visual observations. Building on this idea, the authors introduce `VisPhyBench`, a benchmark of 209 scenes derived from 108 physical templates, together with a multi-view evaluation protocol covering semantic reconstruction, motion consistency, and overall judged quality. The system also includes code repair and fallback mechanisms to improve executability. The reported experiments suggest that current MLLMs can often recover semantic scene content better than underlying physical parameters or consistent dynamics.

**Compliance With Llm Reviewing Policy:**

Affirmed.

**Final Justification:**

The authors have comprehensively addressed all weaknesses and questions with additional experiments and analyses, including cross‑judge evaluations, parameter accuracy measurements, and robustness checks without auxiliary context.

All concerns are sufficiently clarified, and the core conclusions are validated to be robust across different evaluation setups.

I maintain my original recommendation of weak accept.

**Key Questions For Authors:**

1. How do you separate physical reasoning ability from code generation skill and familiarity with the simulation backend?
2. How do you account for cases in which the underlying dynamics are not uniquely identifiable from the provided observations?
3. How robust are the conclusions to removing auxiliary context or replacing the current judge model?

**Limitations:**

Yes. The paper already acknowledges several important limitations, including the use of synthetic rigid-body scenarios and the limited evidence for real-world generalization. It may also be worth stating more explicitly that the benchmark currently mixes physical reasoning with program synthesis ability.

**Strengths And Weaknesses:**

Strengths:
- The framing is original and offers a distinctive alternative to QA-style physical reasoning evaluation.
- The task requires the model to externalize its hypothesis in a testable form.
- The benchmark and evaluation protocol are reasonably rich and well thought out.
- The paper includes comparisons across models, code backends, and generation settings.
- The limitations and implementation details are presented with reasonable clarity.

Weaknesses:
- The benchmark still conflates physical reasoning with code generation ability and backend familiarity.
- With only sparse observations, some physical parameters may be underdetermined.
- Some parts of the evaluation still depend on LLM-based judgment.
- The fallback mechanism makes executability easier to achieve, but may blur the distinction between valid execution and physically correct reconstruction.
- The current scenarios remain relatively simple and synthetic.

---

> ### Author Rebuttal · Authors · 2026-03-31
>
> ## For Weakness & Question
>
> **W1 & Q1**: The benchmark…
>
> Our goal is not to separate physical reasoning from coding, but to test whether models can express physical understanding through executable simulation code. Since many MLLMs cannot directly generate videos, code provides a practical evaluation interface. To reduce toolchain bias, we evaluate multiple backends and also include direct video-generation baselines such as Sora-2 and Veo-3.1.
>
> **W2 & Q2**: With only…
>
> We agree that two frames are often insufficient to uniquely determine the full dynamics. VisPhyBench is therefore not a strict system-identification task, but a controlled setting for testing whether a model can generate a physically plausible, rollout-aligned future from sparse observations, consistent with prior work on intuitive physics and generative-video evaluation [3-6].
>
> **W3**: Some parts…
>
> Thank you for this question. To reduce reliance on LLM-based judgment, we further evaluate the models with VideoScore2, an automatic multi-dimensional video quality scorer.
>
> |Model|Temporal|Physical|
> |---|---:|---:|
> |Claude-Sonnet-4.5|2.20|2.12|
> |GPT-5|2.23|2.17|
> |Gemini-3-Pro|2.27|2.14|
>
> The results are broadly consistent with ours: temporal consistency aligns directionally with RAFT-EPE, and physical consistency shows the same top-tier grouping. This suggests that our conclusions do not depend on a single LLM judge.
>
>
> We also conduct a direct analysis to assess the physical parameter correctness via MAE between GT and predicted parameters.
>
> |Parameter|GT mean|GPT-5 MAE|Gemini-3-Pro MAE|Claude-S4-5 MAE|
> |---|---:|---:|---:|---:|
> |Gravity|9.82 m/s²|0|0|0|
> |Restitution|0.15|0.17±0.14|0.18±0.13|0.16±0.05|
> |Friction|0.40|0.10±0.07|0.11±0.07|0.10±0.03|
> |Dynamic mass|1.25 kg|0.15±0.17|0.59±0.88|0.26±0.42|
> |Init speed|2.59 m/s|2.07±2.69|1.69±2.42|2.39±3.23|
>
> This reinforces our core claim that current models can generate visually plausible videos yet still fail to recover the quantitative physical parameters governing the dynamics.
>
> **W4**: The fallback ….
>
> Our no-retry success rates are already high: 98% for Three.js and 85% for p5.js. The fallback mechanism is designed only to improve executability, not reconstruction quality. It retries on syntax errors and runtime crashes, but does not change the physical logic, object placement, or simulation parameters. As a result, a video can be executable yet still score poorly on perceptual, motion, and holistic physical-plausibility metrics. In our benchmark, executability is only a prerequisite for evaluation, while reconstruction quality is measured separately.
>
> **W5**: The current…
>
> We agree that VisPhyBench currently focuses mainly on controlled synthetic rigid-body scenes. We view this benchmark as an early exploration of this problem setting, which is also consistent with prior works such as CoPhy [1] and PhyWorld [2], which use relatively simple and controlled scenes.Current MLLMs often cannot reliably identify object types, positions, and layouts, making it difficult to separate perceptual failures from physical reasoning failures.
>
> **Q3**: How robust…
>
> Thank you for this question. Without detection context, all tested models fall to similarly low holistic scores, suggesting that the context mainly helps object discovery and initialization rather than physical reasoning.
>
> We also replace the original Gemini judge with GPT-5.4 and Qwen3-VL-Plus, and further conduct human evaluation. Although exact rankings vary, the overall conclusions remain stable: inter-judge correlations are significant, weak baselines such as SVD-img2vid consistently rank at the bottom. This supports the robustness of our conclusions to both auxiliary context removal and judge replacement.
>
> 1. Different judge models:
>
> |Model|Gemini score|GPT-5.4 score|Qwen3-VL score|
> |---|---:|---:|---:|
> |Veo-3.1|2.56|5.09|7.82|
> |Gemini-3-Pro|3.81|3.99|6.74|
> |GPT-5|3.53|3.33|7.12|
> |GPT-4.1|3.04|3.33|6.84|
> |Claude-Sonnet-4.5|2.50|3.60|6.52|
> |Qwen3-VL-Plus|2.13|2.25|4.94|
> |SVD-img2vid|1.44|1.42|2.92|
>
>
> 2. Sample-level judge agreement correlation table(all p<0.001):
>
> |Judge Pair|Pearson r|Spearman ρ|
> |---|---:|---:|
> |Gemini vs GPT-5.4|0.41|0.37|
> |Gemini vs Qwen3-VL|0.40|0.43|
> |GPT-5.4 vs Qwen3-VL|0.65|0.70|
>
> 3. Model-level scores without detection context
>
> |Model|Mode|LPIPS↓|CLIP-Img↑|DINO↑|CLIP-Cap↑|BERTScore↑|RAFT-EPE↓|Gemini↑|
> |---|---|---:|---:|---:|---:|---:|---:|---:|
> |GPT-5|with detection|0.17|0.89|0.86|0.26|0.84|33.65|3.50|
> |GPT-5|no detection|0.29|0.88|0.81|0.28|0.85|30.55|1.98|
> |Gemini-3-Pro|with detection|0.14|0.90|0.84|0.26|0.85|36.20|3.80|
> |Gemini-3-Pro|no detection|0.21|0.90|0.84|0.25|0.85|31.77|1.84|
> |Claude Sonnet 4.5|with detection|0.16|0.90|0.83|0.26|0.85|36.20|2.39|
> |Claude Sonnet 4.5|no detection|0.27|0.89|0.82|0.25|0.86|31.80|1.65|
>
> **Note:** Due to the rebuttal word limit, we will provide additional details in the discussion phase if needed.
>
> **References:** See anonymous.4open.science/r/VisPhyWorld-rebuttal-B6BF/README.md

---

> > ### Author Rebuttal · Reviewer_mREd · 2026-04-04
> >
> > I thank the authors for their detailed responses and the effort put into the revision. After careful consideration, I have decided to maintain my original score.

---

> > > ### Author Response · Authors · 2026-04-07
> > >
> > > We sincerely thank the reviewer for the constructive feedback and for recognizing the value of our work.

---

### Official Review · Reviewer_JK2M · 2026-03-18

**Soundness:** 3
**Presentation:** 3
**Significance:** 2
**Originality:** 3
**Overall Recommendation:** 4
**Confidence:** 4

**Summary:**

This paper proposes VisPhyWorld, an evaluation framework for physical reasoning in multimodal LLMs where the model must convert visual observations into executable simulation code rather than answer VQA-style questions. Given two key frames (and optionally object detections), the model outputs a motion analysis, a structured scene specification, executable code, and a rendered rollout video. The paper also introduces VisPhyBench, a benchmark with 209 scenes derived from 108 physical templates spanning both 2D and 3D settings, and evaluates several frontier MLLMs together with pixel-space video baselines. The central claim is that current models are relatively strong at semantic scene parsing but much weaker at recovering physical parameters and dynamics well enough to faithfully re-simulate the scene.

**Compliance With Llm Reviewing Policy:**

Affirmed.

**Final Justification:**

I've raised my score to WA after checking the rebuttal.

**Key Questions For Authors:**

1. How much does performance change without the detection context? Is the benchmark primarily testing physical reasoning, or is a meaningful part of the gain coming from better object discovery and initialization?
2. Since two frames often underdetermine the true underlying dynamics, how does the benchmark handle multiple physically-valid hypotheses that could produce similar future rollouts?
3. Can the authors add a more direct evaluation of parameter recovery, such as mass, friction, restitution, and initial velocity, not just rendered-video agreement?
4. How fair are the pixel-space baselines in conditioning and prompting, and could stronger world-model or video-generation baselines be included?
5. How well does the framework extend to real videos, longer horizons, cluttered scenes, and richer 3D engines like in BlenderGym[1] or VIGA[2]?

[1]: Benchmarking Foundational Model Systems for Graphics Editing, Gu et al. https://arxiv.org/abs/2504.01786
[2]: Vision-as-Inverse-Graphics Agent via Interleaved Multimodal Reasoning, Yin et al. https://arxiv.org/abs/2601.11109

**Limitations:**

Yes

**Strengths And Weaknesses:**

### Strengths
- Strong and original evaluation framing. Requiring executable code makes the model’s hypothesis inspectable, editable, and falsifiable, which is more diagnostic than answer-only physical reasoning benchmarks.
- Good benchmark design. The paper includes both 2D and 3D scenes, multiple model backbones, multiple code/rendering backends, and a multi-metric evaluation suite that separates perceptual fidelity, semantic consistency, text-video consistency, motion plausibility, and holistic quality.
- Useful diagnostic findings. The experiments support an important result: semantic alignment and physical correctness can dissociate, and pixel-space generation can look plausible while still failing on actual dynamics.
- Practical robustness analysis. The paper studies engine choice and retry/self-repair, which makes the evaluation pipeline more realistic and informative.


### Weaknesses
- Limited scope and external validity. The benchmark is based on synthetic, simulator-driven, controlled scenes with relatively simple rigid-body interactions. The conclusions may not transfer cleanly to in-the-wild videos, long-horizon dynamics, cluttered scenes, or more complex 3D environments.
- Potential confound between physics reasoning and code or simulator familiarity. Success depends not only on inferring motion, but also on producing valid code in specific toolchains such as Three.js, P5.js, and Cannon.js. This may partially measure simulator API competence rather than pure physical reasoning.
- Evaluation is still indirect in places. The benchmark measures whether rollouts match visually and physically at the video pixel level, but it is less direct about whether the model recovered the correct latent physical parameters such as mass, friction, restitution, or initial velocity.

---

> ### Author Rebuttal · Authors · 2026-03-31
>
> ## For Weakness
>
> **W1**: Limited scope and external validity.
>
> We agree that VisPhyBench currently focuses mainly on controlled synthetic rigid-body scenes. We view this benchmark as an early exploration of this problem setting, which is also consistent with prior works such as CoPhy [1] and PhyWorld [2], which use relatively simple and controlled scenes.Current MLLMs often cannot reliably identify object types, positions, and layouts, making it difficult to separate perceptual failures from physical reasoning failures.
>
> **W2**: Potential confound between physical reasoning and simulator familiarity.
>
> Our goal is not to separate physical reasoning from coding, but to test whether models can express physical understanding through executable simulation code. Since many MLLMs cannot directly generate videos, code provides a practical evaluation interface. To reduce toolchain bias, we evaluate multiple backends and also include direct video-generation baselines such as Sora-2 and Veo-3.1.
>
> **W3 & Q3**：Evaluation is still indirect in places.
>
> Thank you for the suggestion. We additionally conduct a direct analysis to assess the physical parameter correctness via MAE between GT and predicted parameters.
>
> |Parameter|GT mean|GPT-5 MAE|Gemini-3-Pro MAE|Claude-S4-5 MAE|
> |---|---:|---:|---:|---:|
> |Gravity|9.82 m/s²|0|0|0|
> |Restitution|0.15|0.17±0.14|0.18±0.13|0.16±0.05|
> |Friction|0.40|0.10±0.07|0.11±0.07|0.10±0.03|
> |Dynamic mass|1.25 kg|0.15±0.17|0.59±0.88|0.26±0.42|
> |Init speed|2.59 m/s|2.07±2.69|1.69±2.42|2.39±3.23|
>
> These results are consistent with our other metrics: although Gemini-3-Pro shows strong visual reconstruction, it still makes large errors on dynamic mass and initial speed. This reinforces our core claim that current models can generate visually plausible videos yet still fail to recover the quantitative physical parameters governing the dynamics.
>
> ## For Question
>
> **Q1**: Ablation on detection context.
>
> Thank you for highlighting the need for an ablation on detection context. We evaluated all models again without the detection context.
>
> |Model|Mode|LPIPS↓|CLIP-Img↑|DINO↑|CLIP-Cap↑|BERTScore↑|RAFT-EPE↓|Gemini↑|
> |---|---|---:|---:|---:|---:|---:|---:|---:|
> |GPT-5|with detection|0.17|0.89|0.86|0.26|0.84|33.65|3.50|
> |GPT-5|no detection|0.29|0.88|0.81|0.28|0.85|30.55|1.98|
> |Gemini-3-Pro|with detection|0.14|0.90|0.84|0.26|0.85|36.20|3.80|
> |Gemini-3-Pro|no detection|0.21|0.90|0.84|0.25|0.85|31.77|1.84|
> |Claude Sonnet 4.5|with detection|0.16|0.90|0.83|0.26|0.85|36.20|2.39|
> |Claude Sonnet 4.5|no detection|0.27|0.89|0.82|0.25|0.86|31.80|1.65|
>
> Removing the detection context causes a clear drop in LPIPS and Gemini score across all models, while the other metrics change much less, suggesting that it mainly helps object discovery and initialization rather than directly solving dynamics reasoning. We therefore include detection context to reduce the confound from object localization.
>
> **Q2**: Non-unique Rollouts.
>
> We agree that two frames are often insufficient to uniquely determine the full dynamics. VisPhyBench is therefore not a strict system-identification task, but a controlled setting for testing whether a model can generate a physically plausible, rollout-aligned future from sparse observations, consistent with prior work on intuitive physics and generative-video evaluation [3-6].
>
> **Q3**: Please refer to **W3**
>
> **Q4**: More baselines.
>
> we further included Cosmos and Sora-2:
>
> |Model|LPIPS↓|CLIP-Img↑|DINO↑|CLIP-Cap↑|BERTScore-F1↑|RAFT-EPE↓|Gemini↑|Human Eval↑|
> |---|---:|---:|---:|---:|---:|---:|---:|---:|
> |Sora-2|0.20|0.87|0.87|0.26|--|34.91|2.35|2.83|
> |Cosmos-Predict2.5-2B†|0.52|0.63|0.54|--|--|30.91|1.16|1.16|
>
> All baselines use the same input images and prompt for fairness. We include Cosmos and Sora-2: Sora-2 still lag in physical correctness, while Cosmos performs much worse overall; in human evaluation, we also observe that Cosmos often fails to maintain recognizable objects in later frames. These results suggest even with frontier models, lacks the transparency and controllability of code-based simulation for evaluating physical reasoning.
>
> **Q5**: How well does…
>
> We note that BlenderGym and VIGA target different tasks, focusing on image-level or static-scene reconstruction, whereas VisPhyWorld studies dynamic physical prediction with motion and evaluation. Extending to real videos is important but still difficult, as in our Blender and Unreal experiments, current MLLMs could not reliably generate and repair simulation code within a small fixed number of calls to produce stable, plausible videos. For real-world and cluttered videos, the bottleneck appears even earlier.
>
> **Note:** Due to the rebuttal word limit, we will provide additional details in the discussion phase if needed.
>
> **References:** See anonymous.4open.science/r/VisPhyWorld-rebuttal-B6BF/README.md

---

> > ### Author Rebuttal · Reviewer_JK2M · 2026-04-06
> >
> > My concerns have been adequately addressed.

---

> > > ### Author Response · Authors · 2026-04-07
> > >
> > > We sincerely thank the reviewer for the positive update and for recognizing the value of our work. We are glad that our clarifications on the benchmark scope and the role of detection context, together with the added experiments on parameter analysis, detection-context ablation, and additional video baselines, addressed the initial concerns. We believe these results further support the importance of VisPhyBench as a controlled benchmark for evaluating physical understanding in MLLMs.

---

### Decision · Program_Chairs · 2026-04-30

**Decision:**

Reject

**Comment:**

This paper was reviewed by four experts in the field. The recommendations are (Weak Reject, Weak Accept x 3). The reviewers agree that the paper presents an original evaluation framework for physical reasoning in multimodal large language models through executable simulation code generation, and that the empirical study provides useful diagnostic insights. However, the Reviewers like mDvZ also raise substantial concerns regarding the limited scale and synthetic nature of the benchmark, the possible conflation of physical reasoning with code generation ability and simulator familiarity, and the indirectness of several evaluation metrics. Additional concerns include the reliability of the LLM/VLM-based judge and the limited evidence for broader generalization beyond the current benchmark setting. The authors provided a rebuttal and additional analyses, which addressed some questions. However, after the rebuttal and discussion period, the main concerns regarding benchmark scope, evaluation methodology, and the strength of the paper’s broader claims were not fully resolved. Taking these concerns into consideration, the paper would not be accepted at this time. The authors are encouraged to further strengthen the benchmark and evaluation, clarify the scope of the claims, and improve the manuscript before submitting the work next time or elsewhere.